# Neural Mutual Information Estimation with Reference Distributions

## Abstract

Estimating mutual information (MI) from data is a fundamental task in machine learning and data science, yet it remains highly challenging even with state-of-the-art estimators. This work proposes a new distribution-free MI estimator based on reference distributions. Unlike existing works that only discern between the joint distribution and the marginal distribution, which can easily overfit in high-MI settings, our method compares them with extra reference distributions. These artificial distributions share the same marginals as the original distributions but have known dependence structures, providing additional signals for more accurate dependency modelling. Experiments on synthetic tasks with non-Gaussian, high-dimensional data and real-world applications including Bayesian experimental design and self-supervised learning demonstrate the potential of our approach.

## 1 Introduction

Mutual information (MI), defined as the Kullback-Leibler divergence between the joint probability and the product of marginals, is a fundamental measure of the dependence between random variables. Unlike correlation, MI captures non-linear statistical dependencies between high-dimensional variables, making it a robust measure of dependence in diverse cases [1]. Consequently, MI has been widely applied across diverse fields, such as representation learning [2; 3; 4], Bayesian experimental design [5; 6], information bottleneck [7; 8], domain generalization [9; 10], and causality [11; 12].

Despite its widespread applications, estimating MI in high-dimensional spaces from raw data is highly challenging. Early methods work in a non-parametric fashion, utilizing techniques like binning or kernel density estimation [13; 14; 15; 16; 17]. As non-parameteric methods typically do not scale well with sample size or data dimensionality [18], recent studies have focused on utilizing powerful parametric models such as deep neural networks in MI estimation [19; 2; 20; 21; 22; 23; 24; 25]. In these estimators, a neural network (often known as the critic) is trained to discern or classify samples from the joint distribution and the marginal distribution. Upon convergence, it can be shown that the learned network corresponds to an estimate of the density ratio between the joint distribution and the product of marginals [19; 2; 20; 21; 22], or that the loss function of these estimators directly optimize a variational lower bound of MI [19; 2], thereby offering an estimate of MI. Compared to non-parametric methods, estimating MI through neural network training not only handles large sample sizes more effectively but also scales better with high-dimensional data. Thereby, it is increasingly gaining popularity in both machine learning and statistics communities.

Although neural network-based MI estimators have achieved success in numerous tasks, they are recently found to suffer from the so-called *high-discrepancy issue*, particularly when the underlying mutual information is high or the data dimensionality is high. In such scenarios, distinguishing samples from the joint and the marginal distributions becomes considerably easier, leading the network to easily overfit with limited data. This curse of high-discrepancy has been empirically observed in multiple applications [26; 22; 27] and has also been formally analyzed in theoretical studies [28; 29].

In this work, we propose Mutual Information Multinomial Estimation (MIME), a novel approach tailored for estimating mutual information accurately in high-discrepancy settings, drawing upon recent advances in multi-classes ratio estimation [22] and vector copula theory [30]. Unlike many existing methods that solely focus on discerning samples between the joint and the marginal distributions, which can easily overfit in high-discrepancy scenarios, MIME further compares them with

a set of reference distributions whose marginal distributions are the same as the original data distribution but exhibit various joint dependency structures. These additional comparisons encourage the network to learn a useful representation that captures the small differences in dependence structures, thereby effectively alleviating overfitting. To sum up, our main contributions are as follows:

- We propose a new mutual information estimator tailored for high-MI cases, combining the strengths of existing generative and distribution-free methods while avoiding their weaknesses;

- We systematically compare our estimator against state-of-the-art neural estimators on not only high-dimensional, non-Gaussian synthetic dataset with known MI but also real-world applications.

## 2 BACKGROUND

The mutual information (MI) between variables $X$ and $Y$ is defined as the Kullback-Leibler (KL) divergence between the joint distribution $p(x, y)$ and the product of the marginal $p(x)p(y)$

$$I(X; Y) = KL[p(x, y)\|p(x)p(y)] = \mathbb{E}\left[\log \frac{p(x, y)}{p(x)p(y)}\right]. \tag{1}$$

Stronger dependence between $X$ and $Y$ leads to larger divergence between $p(x, y)$ and $p(x)p(y)$ and thereby a high MI. Given a dataset $\mathcal{D} = \{x^{(i)}, y^{(i)}\}_{i=1}^n$ drawn from the joint distribution $p(x, y)$, our goal in this work is to estimate $I(X; Y)$ from the empirical samples $\mathcal{D}$.

**Distribution-free MI estimator**. Many methods have been developed to estimate (1) from $\mathcal{D}$ without making any assumptions about the underlying probability distributions. One popular method is to utilize the Donsker-Varadhan (DV) representation of KL divergence [31]:

$$I(X; Y) \geq \sup_{f \in \mathcal{F}} \mathbb{E}_{p(x,y)}[f(x, y)] - \log \mathbb{E}_{p(x)p(y)}[e^{f(x,y)}] =: \hat{I}_{\mathrm{DV}}(X; Y), \tag{2}$$

where $\mathcal{F}$ can be any class of function $\mathcal{F} : \mathcal{X} \to \mathbb{R}$ satisfying the integrability constraints of the theorem. A typical choice of $\mathcal{F}$ is the class of deep neural network. In such case, MI estimate reduces to training a neural network $f$ to maximize the objective (2). Similarly, the NWJ estimator [32] makes use of the $f$-divergence lower bound to estimate MI: $I(X; Y) \geq \sup_{f \in \mathcal{F}} \mathbb{E}_{p(x,y)}[f(x, y)] - \mathbb{E}_{p(x)p(y)}[e^{f(x,y)-1}] =: \hat{I}_{\mathrm{NWJ}}(X; Y)$. Both DV and NWJ estimators leads to a lower bound of MI. Note that the equalities in the two bounds hold if $\mathcal{F}$ is wide enough.

Alternatively, one can estimate MI by employing techniques from density ratio estimate [33; 20; 22]:

$$I(X; Y) \approx \mathbb{E}_{p(x,y)}[\log \hat{r}(x, y)] =: \hat{I}_{\mathrm{ratio}}(X; Y) \tag{3}$$

where $\hat{r}(x, y) \approx p(x, y)/p(x)p(y)$ is the estimated density ratio. To learn $r$, one can adopt a process similar to GAN training [34; 35], where a classifier is trained to distinguish samples from the joint distribution $p(x, y)$ and samples from the marginal distribution $p(x)p(y)$. It can be shown that upon convergence, the output of the classifier estimates the log density ratio. This type of estimator is found more robust than DV-like estimators [3]; however, the MI estimated is not a lower bound.

It is worth noting that these two lines of methods are closely related to each other. On one hand, it can be shown that the optimal $f$ in DV and NWJ estimators is, up to some constant, the log density ratio $\log p(x, y)/p(x)p(y)$. Conversely, the estimated ratio $\hat{r}$ in ratio-estimating method can in turn be used in the DV representation to obtain a lower bound estimate of MI. Regarding the learning process, ratio-estimating method can be seen as optimizing a lower bound of the Jensen-Shannon divergence between $p(x, y)$ and $p(x)p(y)$, whereas DV and NWJ methods optimize KL divergence.

**Curse of high discrepancy**. Despite their advantages over non-parametric methods, distribution-free estimators often suffer from the so-called high-discrepancy issue in practice. High-discrepancy issue typically happens when the dimensionality of data is high and/or the underlying MI is high, which results in a large KL between $p(x, y)$ and $p(x)p(y)$. However, it is shown that asymptotically, any distribution-free estimator of the KL divergence between two distributions $p$ and $q$ will inevitably suffer from an estimation variance that is exponential to the ground truth value [29; 28]:

$$\mathbb{V}[\hat{KL}[p\|q]] \approx O(e^{KL[p\|q]}) \tag{4}$$

which implies that methods based on KL divergence estimation (e.g. DV, NWJ) will be impractical

if the underlying MI is large or the intrinsic dimensionality of the data is high, both of which can lead to a large KL. The situation becomes even worse when we learn estimators with mini-batches. This issue raises concerns about the reliability of DV- and NWJ-like estimators in high-MI settings.

We highlight that the impact of the high discrepancy issue is beyond KL divergence-based estimators. Recent works have confirmed that high-discrepancy issue also exists in ratio-based estimator (3), where the classifier in the estimator tends to overfit when samples $x, y \sim p(x, y)$ and samples $x, y \sim p(x)p(y)$ are overly easy to discern [21; 36; 27; 22; 37]. One intuitive explanation is that when the gap between $p(x, y)$ and $p(x)p(y)$ is large, there exist many classifiers that can achieve almost perfect accuracy but have differently estimated ratios, leading to inaccurate estimate of MI.

## 3 METHOD

In this work, we propose a new MI estimator aiming to address the aforementioned high-discrepancy issue in distribution-free MI estimation. The key idea is, unlike existing works that solely compare the joint distribution $p(x, y)$ with the product of the marginal distributions $p(x)p(y)$, which easily causes overfitting, we also compare them with additional reference distributions which share the same marginals as the original distribution but have various dependence structures. These additional comparisons provide more fine-grained signals for accurately modelling the dependence structures of $p(x, y)$ and $p(x)p(y)$, allowing us to better model the dependence between $X$ and $Y$.

**Reference-based MI estimation**. The core of our method is to formulate MI estimation as a multinomial classification problem, where we not only distinguish samples from $p(x, y)$, $p(x)p(y)$ but also samples from specific-designed reference distributions. Specifically, consider classifying samples $x, y$ from four distributions $p_1(x, y), p_2(x, y), p_3(x, y), p_4(x, y)$ with a classifier $h$. This can be done by training $h$ with the following objective function:

$$\max_h \ \mathcal{L}(h) = \mathbb{E}_{p(c)p_c(x,y)} \Big[ \log \frac{e^{h_c(x,y)}}{\sum_{k=1}^4 e^{h_k(x,y)}} \Big], \tag{5}$$

$$p_1(x, y) = p(x, y), \qquad p_2(x, y) = q(x, y), \qquad p_3(x, y) = q(x)q(y), \qquad p_4(x, y) = p(x)p(y).$$

where $h_c$ denotes the logit (i.e. the unnormalized class probability) computed for class $c$ and $q(x, y)$ is some reference distribution with marginals $q(x), q(y)$. This reference distribution is a generative model learned from the data, which can be as simple as a Gaussian distribution that approximates $p(x, y)$ or can be as flexible as a diffusion model trained on the data [38]. A good choice of reference should make use of the inductive bias about the problem (MI estimation), as we discuss later.

Upon convergence, it can be shown that when the prior $p(c)$ is uniform, the optimal $h$ satisfies [22]:

$$\log \frac{p_i(x, y)}{p_j(x, y)} = h_i(x, y) - h_j(x, y), \tag{6}$$

thus allowing the mutual information to be estimated as:

$$I(X; Y) \approx \hat{I}_h(X; Y) := \mathbb{E}_{p(x,y)} \Big[ h_1(x, y) - h_4(x, y) \Big], \tag{7}$$

where the expectation on the RHS can be readily approximated by Monte Carlo integration. Note that this estimator is neither a lower or upper bound estimate[1].

The whole procedure for our reference-based MI estimator therefore include two independent steps:

- *Reference construction*. In this first step, we learn the reference $q(x, y)$ with data $(x, y) \sim p(x, y)$.
- *Classifier training*. This corresponds to training the classifier $h$ using objective (5) given $q(x, y)$.

which is highly modular and can be implemented with any reference distributions and any classifiers. In this procedure, the introduction of $p_2(x, y)$ and $p_3(x, y)$ does not affect the theoretical correctness of the ratio estimator (6); instead, they function as implicit regularization to alleviate overfitting.

---

[1] Alternatively, one can use the DV representation (2) to estimate MI by setting $f(x, y)$ in (2) to be the estimated ratio: $f(x, y) = h_1(x, y) - h_4(x, y) \approx \log p(x, y)/p(x)p(y)$. This guarantees that the estimated MI is a lower bound of $I(X; Y)$, however it needs to obtain samples from $p(x)p(y)$, which is unneeded in (7).

This estimation strategy can be seen as a new paradigm for combining discriminative (distribution-free) methods [39; 21; 33; 22] and generative modelling [40; 30] for accurate MI estimation.

**Choices of reference distributions**. We next discuss the design of reference distributions. While any distributions can be used in principle, we focus on those that are *marginal preserving*, i.e. the marginals $q(x)$ and $q(y)$ of the reference distribution $q(x, y)$ should closely approximate those of the original distribution: $q(x) \approx p(x), q(y) \approx p(y)$. This property is motivated by the fact that MI is irrelevant to marginals, so a good reference should only differ from $p(x, y)$ in dependence structure; When asking the network $h$ to distinguish among the various distributions in the above framework, these marginal-preserving references enforce the network to learning a useful representation to distinguish the small changes in dependence structures, leading to accurate MI estimate.

To this end, we design the reference distributions to be *vector Gaussian copula* (VGC) models [30]. It can be seen as either a degenerated case of a standard flow model [40; 41] or a generalization of the classic Gaussian copula model. Specifically, a VGC $q(x, y)$ is implicitly defined by its data generation process, where data $x, y \sim q_($ $x, y)$ can be seen as generating from the following process:

$$x = f(\epsilon_{\leq d}), \quad y = g(\epsilon_{>d})$$
$$\epsilon \sim \mathcal{N}(\epsilon; 0, \Sigma) \tag{8}$$

where $f : \mathbb{R}^d \to \mathbb{R}^d$ and $g : \mathbb{R}^{d'} \to \mathbb{R}^{d'}$ are two bijective functions and $\Sigma \in \mathbb{R}^{(d+d') \times (d+d')}$ is a p.s.d matrix where $\Sigma_{ii} = 1, \forall i$. Here $\epsilon_{\leq d}$ and $\epsilon_{>d}$ are the first $d$ and the last $d'$ dimensions of $\epsilon$. Note that this model degenerates to a Gaussian copula when $f$ and $g$ are element-wise functions.

We implement the two bijective functions $f$ and $g$ by flow-based models e.g. [40; 41]. The parameters $\{f, g, \Sigma\}$ of VGC can then be learned by first training the flows $f$ and $g$ with data $x \sim p(x)$ and $y \sim p(y)$ respectively, then learning $\Sigma$ as $\Sigma = \mathbb{E}[\epsilon \epsilon^\top]$ where $\epsilon_{\leq d} = f^{-1}(x)$ and $\epsilon_{>d} = g^{-1}(y)$.

We take the two reference distributions in (5) as two VGCs that approximates $p(x, y)$ and $p(x)p(y)$ respectively. We highlight the following two useful properties of VGC as reference distributions:

(a) *Reasonably close to $p(x, y)$ or $p(x)p(y)$.* The marginal distributions of a well-trained VGC are almost the same as those of $p(x, y)$ and $p(x)p(y)$, with its dependence structure a Gaussian approximation to that of $p(x, y)$ or $p(x)p(y)$, thereby is reasonably close to $p(x, y)$ or $p(x)p(y)^2$;

(b) *Easy to sample.* Sampling from a VGC can readily be done by (8). This allows us to generate arbitrarily many samples from the reference distributions when learning the network with (5).

In the theoretical analysis below, we analyze how these two properties contribute to an accurate estimate of MI. In the appendix, we further compare this reference with other choices of references.

## 4 THEORETIC ANALYSIS

In practice, we replace the expectation in (5) with $N$ samples to learn our MIME estimator via: $\hat{h} = \arg\max_h \mathcal{L}^N(h)$. After training, we estimate $I(X; Y)$ via Monte Carlo estimation: $\hat{I}_{\hat{h}}^N(X; Y) := \frac{1}{N} \sum_{i=1}^N [\hat{h}_1(x_i, y_i) - \hat{h}_4(x_i, y_i)]$. In Proposition 1, we prove that our MIME estimator is consistent.

**Proposition 1** (Consistency of reference-based MI estimate). *Assuming that the classifier $h_c : X \times Y \to \mathbb{R}$ is uniformly bounded. For every $\varepsilon > 0$, there exists $N(\varepsilon) \in \mathbb{N}$, such that*

$$\left| \hat{I}_{\hat{h}}^N(X; Y) - I(X; Y) \right| < \varepsilon, \forall N \geq N(\varepsilon), a.s..$$

*Proof.* See Appendix A. This result is an extension of a result in [22]. □

This confirms that the probability that the estimated MI of MIME becomes arbitrarily close to the true MI converges to one, as the amount of data used in the proposed estimator approaches infinity.

Furthermore, we have the following formal result about the advantages of our proposed estimator.

---

[2] According to recent copula theory [30], the difference between two distributions $p(x, y)$ and $q(x, y)$ can be factorized into the difference in the (multivariate) marginals and the difference in the dependence structure.

**Proposition 2** (Controlled error in reference-based MI estimate). *Define* $\log \hat{r}_{i,j}(x,y) := h_i(x,y) - h_j(x,y)$ *with* $h$ *being the classifier defined as above. Upon convergence, we have that:*

$$\left| \log \hat{r}_{1,4}(x,y) - \log \frac{p(x,y)}{p(x)p(y)} \right| \leq 3\sup_i \left| \log \hat{r}_{i,i+1}(x,y) - \log \frac{p_i(x,y)}{p_{i+1}(x,y)} \right|,$$

*Proof.* See Appendix A. □

Proposition 2 tells that if the ratios $p(x,y)/q(x,y)$, $p(x)p(y)/q(x)q(y)$ and $q(x,y)/q(x)q(y)$ can all be estimated accurately in $h$, then the ratio $p(x,y)/p(x)p(y)$ estimated by the same network will also be accurate. This is exactly the case in our method: the first two ratios can be estimated accurately due to the reduced distributional discrepancies (see property (a)). For the ratio $q(x,y)/q(x)q(y)$, the discrepancy between these two reference distributions may be high, however we can generate infinite samples from them in learning (see property (b)), so this ratio can also be learned accurately.

We note that other choice of reference distributions, such as those in [22], may not enjoy the same benefit as our references. The fundamental reason is that it is difficult to find references that are respectively close enough to $p(x,y)$ and $p(x)p(y)$, but are also close to each other. Our generative references sidestep this challenge by not requiring the two references to have a small discrepancy.

## 5 RELATED WORKS

**Alleviating high-discrepancy issue in statistical divergence estimation**. Several methods have been developed to address the high discrepancy issues in the estimation of statistical divergence the estimation of density ratio. One method is to reduce the estimation variance by clipping the output of the estimator [29], which can be seen as trading low variance with additional bias. Another line of methods solves the problem by the divide-and-conquer principle [21; 42], where the statistical divergence/density ratio between some intermediate distributions are first estimated, followed by an aggregation step to recover the original statistical divergence/density ratio from these intermediate estimations. Due to the reduced distributional discrepancy, each of these intermediate estimates can be expected to be accurate. However, divide-and-conquer methods are found to suffer from the *distribution-shift* issue during the aggregation step, which can lead to estimation inaccuracy. Unlike these methods, our method does not incur extra estimation bias and is free from distribution shift.

**Multi-classes ratio/MI estimate**. Similar to our work, some recent works also utilize a set of reference distributions $q$ to improve density ratio estimate [43; 22], and the use of reference distributions in MI estimation have also been explored in [44; 45]. A crucial difference between our method and these approaches is that the reference distributions in our method are *marginal-preserving* i.e. their marginal distributions $q(x)$ and $q(y)$ are close to the original marginal distributions. As aforementioned, such marginal-preserving property is important as it enforces the network to focus on modelling the changes in dependence structures rather than modelling the marginal distributions, which are irrelevant for MI estimation. This importance will be verified in the experiment section.

**Generative methods for mutual information estimation**. In addition to the above distribution-free methods for MI estimate, there also exist a set of MI estimators [46; 28; 47; 48; 38] based on generative modelling. These methods work by learning a generative model (e.g. flow-based model [40; 41] or diffusion model [38]) to approximate the density or the score of the data distribution rather than comparing between different densities, thereby are free from the high-discrepancy issue. However, the performance of such estimators is dominated by the quality of the learned generative models, and generative modelling as done by e.g. flow-based model is well-known hard in high-dimensional cases [49; 50; 51; 41] and can suffer from model mis-specification. Our method also uses generative models, but only as informed reference distributions, which is less sensitive to model mis-specification and the quality of the learned generative models.

**Vector Gaussian copula for MI estimate**. We discuss a recent work [48] highly relevant to our method. This work first Gaussianizes the two marginals via two distint flow models, then approximates the joint distribution of the (marginally) Gaussianized data by a Gaussian distribution, from which mutual information is calculated. This process can be seen as modelling the data distribution as a vector Gaussian copula model (8) and directly computing MI from this model. This way of MI calculation is accurate if the underlying data distribution can be well-approximated by a vector Gaussian copula, but can be inaccurate in the case of model mis-specification. Rather than fully relying on the vector Gaussian copula approximation, which is risky, we use it as references solely.

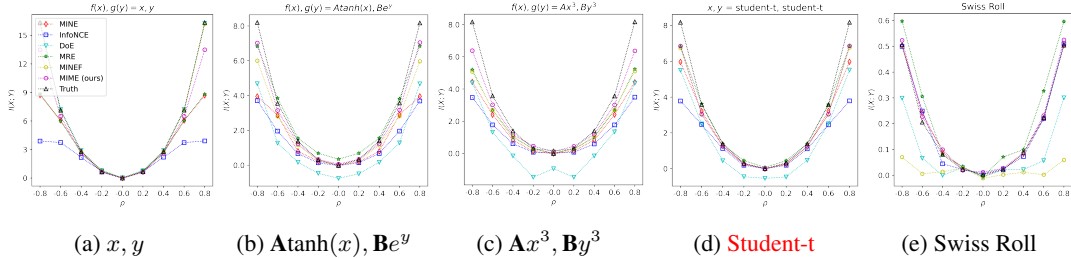

(a) $x, y$      (b) $\mathbf{A}\tanh(x), \mathbf{B}e^y$      (c) $\mathbf{A}x^3, \mathbf{B}y^3$      (d) Student-t      (e) Swiss Roll

Figure 1: Comparison of different MI estimators under different $\rho$ in four representative synthetic datasets. The dimensionalities of the data $X, Y \in \mathbb{R}^d$ in each case are 64, 32, 32, 32, 2 respectively.

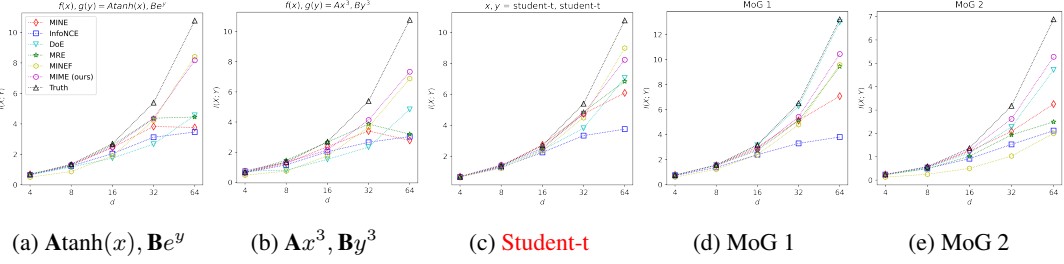

(a) $\mathbf{A}\tanh(x), \mathbf{B}e^y$      (b) $\mathbf{A}x^3, \mathbf{B}y^3$      (c) Student-t      (d) MoG 1      (e) MoG 2

Figure 2: Comparison of different MI estimators under $\rho = 0.7$ and various data dimensionality $d$.

## 6 EXPERIMENT

**Baselines**. We compare our new reference distribution-based MI estimator with five representative baselines in the field: MINE [19], InfoNCE [2], MRE [22], DoE [28] and MIENF [48]. Specifically:

- *Mutual Information Neural Estimate (MINE)*. This method estimates MI by the Donsker-Varadhan representation and a neural network $f$: $I(X;Y) = \sup_f \mathbb{E}_{p(x,y)}[f(x,y)] - \log \mathbb{E}_{p(x)p(y)}[e^{f(x,y)}]$.

- *InfoNCE*. Let $Y = \{y_1, \ldots, y_{N-1}\}$ be $N-1$ negative samples from a proposal distribution, *InfoNCE* estimates MI by learning a neural network $f$ to minimize the loss function $\mathcal{L} = -E_{p(x,y)}\left[\log \frac{f(x,y)}{\sum_{y_i \in Y \cup \{y\}} f(x,y_i)}\right]$, which is upper bounded by MI: $I(X,Y) \geq \log(N) - \mathcal{L}$.

- *Multinomial Ratio Estimate (MRE)*. A state-of-the-art method for density ratio estimation which also works by classifying samples from multiple distributions. The key difference between MRE and our method is the reference distributions in MRE are not marginal-preserving and generative.

- *Difference of Entropy (DoE)*. A generative method which works by first estimating the two entropies $H[Y]$ and $H[X|Y]$ with density estimators $q$, then calculate MI as $I(X;Y) = H[Y] - H[X|Y]$. Here, we study an improved variant of DoE where $q$ is realized as a flow model [40]. This flow model is the same as the flow model in our method, which allows controlled comparison.

- *Mutual Information Estimate via Normalizing Flows (MIENF)*. A state-of-the-art generative method for MI estimation which works by first Gaussianizing the two marginals with two separately-trained flow models, then use a Gaussian distribution to approximate the (marginally) Gaussianized data, which permits a closed-form expression for MI. See Section 4 for more details..

**Neural network architecture and optimizer**. Please refer to Appendix B for the relevant details.

### 6.1 SYNTHETIC DATA

We first investigate the performance of our method on tractable simulated cases, where the mutual information is either analytically known or can be computed numerically up to very high precision.

**Setup**. We consider estimating MI with data simulated from the following three classes of models. These benchmarks are similar to those in state-of-the-art benchmarks [53; 54] and can be seen as an instantiation of the so-called fine distributions [54] for principally evaluating modern MI estimators.

|             | MINE        | InfoNCE     | MIENF       | MRE         | MIME        |
|-------------|-------------|-------------|-------------|-------------|-------------|
| $I(X;Y)=3$  | $2.7\pm0.1$ | $2.2\pm0.1$ | $5.1\pm0.3$ | $4.7\pm0.5$ | $2.5\pm0.2$ |
| $I(X;Y)=7$  | $5.9\pm0.2$ | $3.7\pm0.2$ | $9.8\pm0.4$ | $6.1\pm0.2$ | $5.9\pm0.4$ |
| $I(X;Y)=9$  | $6.2\pm0.2$ | $3.8\pm0.1$ | $11\pm0.4$  | $6.4\pm0.3$ | $6.4\pm0.6$ |

(a) $\hat{I}(X;Y)$

|             | MINE        | InfoNCE     | MIENF       | MRE         | MIME        |
|-------------|-------------|-------------|-------------|-------------|-------------|
| $I(X;Y)=3$  | $2.9\pm0.1$ | $2.4\pm0.1$ | $0.7\pm0.1$ | $4.3\pm0.3$ | $3.2\pm0.2$ |
| $I(X;Y)=7$  | $5.7\pm0.2$ | $3.7\pm0.1$ | $1.4\pm0.2$ | $6.2\pm0.4$ | $6.8\pm0.4$ |
| $I(X;Y)=9$  | $6.3\pm0.2$ | $3.9\pm0.1$ | $2.6\pm0.3$ | $7.6\pm0.3$ | $8.3\pm0.4$ |

(b) $\hat{I}(e(X);e(Y))$

Table 1: Experiments on the image benchmarks from [52], which contain images of rectangles and Gaussian plates. Results shown are for the case of rectangles (results for Gaussian plates are similar; see Appendix B). Left: estimating MI with the original images. Right: estimating MI with the representation of an autoencoder $e : \mathbb{R}^{32\times32} \to \mathbb{R}^8$. Results obtained from 8 independent runs.

- *Nonlinear transformation of multivariate Gaussian.* The first model we consider is a generalized, high-dimensional version of the cubic Gaussian tasks widely used in literature [39; 29; 47]. Data $X \in \mathbb{R}^d, Y \in \mathbb{R}^d$ in this task is generated as follows:

$$x = f(\epsilon_{\leq d}), \qquad y = g(\epsilon_{>d}), \qquad \epsilon \sim \mathcal{N}(\epsilon; \mathbf{0}, \Sigma),$$

where $\Sigma$ is a sparse convariance matrix satisfying $\Sigma_{i,i+d} = \rho, \Sigma_{i,i} = 1$ and $\Sigma_{i,j} = 0$ for any other $j \neq i + d, j \neq i$. $f : \mathbb{R}^d \to \mathbb{R}^d$ and $g : \mathbb{R}^d \to \mathbb{R}^d$ are some bijective functions. A difference between $f, g$ in this work and those in [39; 29; 47; 53] is we further introduce two invertible matrices $\mathbf{A}, \mathbf{B}$ to entangle the dimensions of $X$ and $Y$. Here $\rho$ controls the dependence between $X$ and $Y$. The ground truth MI in this model is given by $I(X;Y) = I(\epsilon_{\leq d}; \epsilon_{>d}) = -\frac{d}{2}\log(1 - \rho^2)$.

- *Mixture of multivariate Gaussians (MoG).* The second case we consider is the mixture of $M$ high-dimensional Gaussian distributions whose likelihood function being is as follows:

$$p(x, y) = \frac{1}{M} \sum_{k}^{M} \mathcal{N}([x, y]; \mu_k, \Sigma_k)$$

where $\Sigma_1...\Sigma_M$ is a set of sparse covariance matrices that are the same as in the non-linear Gaussian model. Note that the dependence factor $\rho_k$ can be different for each $\Sigma_k$. Unlike the non-linear Gaussian model, the MI of this model is not analytically known; however, we can approximate it with Monte Carlo integration due to the availability of both the joint distribution $p(x, y)$ and the marginal distributions $p(x), p(y)$. The variance of MC integration vanishes given sufficient data.

- *Correlated images.* We further consider the benchmarks from [52], which contain correlated images of rectangles and Gaussian plates. This benchmark is selected to investigate the performance of our generative model-based method in high-dimensional settings. Images are with size $32 \times 32$.

We generate a total number of $n = 10^4$ data from each model and estimate MI from the generated data. All results are collected from 8 independent runs and we report their average values. Our test cases include data with Gaussianity, skewness, bounded values, long tails, low-dimensional manifold, high-dimensionality, varying dependence levels and non-Gaussian dependence structures.

**Varying the dependence level** $\rho$. We first investigate whether our method can estimate MI accurately when we vary the dependence level $\rho$. Figure 1 shows the results for the non-linear Gaussian task with five typical choices of non-linear transformations $f, g$. From the figure, it is evident that the proposed estimator consistently ranks within the top two methods across all scenarios, positioning it as a robust method that is widely applicable to different patterns in strcutured data.

**Varying the dimensionality** $d$. In Figures 2, we further compare the performance of the various MI estimators under different data dimensionality. In this task, a higher $d$ leads to a higher MI and thereby a more challenging task of MI estimation. We see that the proposed MIME method clearly scales better w.r.t $d$ than most methods, suggesting that it can better estimate MI in high MI settings.

**High-dimensional image data**. Despite the strong performance on structured data, we observe that our method struggles when applied to high-dimensional image data (see Table 1.a). This is likely due to challenges in training the two high-dimensional flow models in eq.(8) within our method, which results in low-quality reference distributions. Fortunately, this issue can be effectively mitigated by first learning a low-dimensional representation of the data before MI estimation, as suggested in [52; 55]. By doing so, generative modeling becomes considerably easier, leading to high-quality reference distributions. These results highlight the limitations of our method when dealing with high-dimensional image data and the importance of dimensionality reduction methods in these cases.

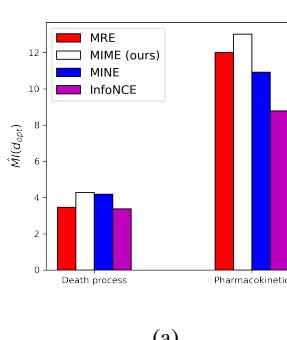 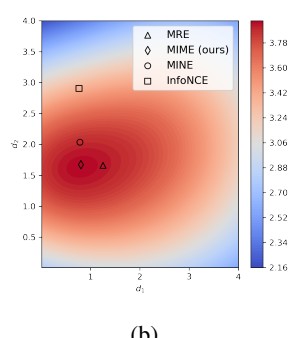 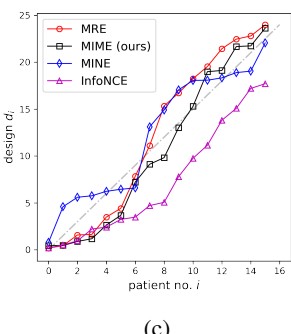

(a)        (b)        (c)

Figure 3: Experiment results for BED. (a) Comparing the utility of the optimal design $\mathbf{d}^*$ found by different MI estimators. (b) Visualizing the contour of the underlying function between the utility and the design $\mathbf{d}$ in the death process task. (c) Visualizing the optimal designs found in the PK task. The results in (a) are collected from 5 independent runs and we report their average values.

## 6.2 BAYESIAN EXPERIMENTAL DESIGN

We next consider the application of our method in Bayesian experimental design (BED), an important task in science and engineering. In BED, we wish to design experiments so that the data collected in the experiment will be maximally informative for understanding the underlying process of the experiment. For example, we may wish to find out the best policy design $\mathbf{d}$ for epidemic disease surveillance and intervention, so that we can infer the properties $\theta$ of the epidemic as accurate as possible with minimal observed data $X$. Such a task can be mathematically described as optimizing the conditional mutual information $I(X; \theta | \mathbf{d})$ between the experiment outcomes $X$ (e.g. the observed dynamics of the infectious people in the population) and the properties of interest $\theta$ (e.g. the infectious rate of the epidemic) w.r.t the experiment design $\mathbf{d}$ (e.g. the intervention policy):

$$\max_{\mathbf{d}} I(X; \theta | \mathbf{d})$$

**Setup**. Many methods have been developed for finding the optimal design in BED [56; 6; 57; 5; 58; 59]. Here, we focus on the most general setup that the data generating process in the experiment is a black-box i.e. we know nothing about the details of the underlying data generation process but only the outcomes and the experiment design used. To find the optimal design $\mathbf{d}^*$ in such case, we use Bayesian optimization (BO) [60; 61] as in existing works to optimize $I(X; \theta | \mathbf{d})$ w.r.t the design $\mathbf{d}$. In this work, we use a total number of $T = 30$ BO steps to find the optimal design $\mathbf{d}^*$, where in each step we run $n = 3000$ experiments under design $\mathbf{d}$ to collect data $\mathcal{D} = \{x^{(i)}, \theta^{(i)}\}_{i=1}^n$ from the experiment. This data is then used to estimate $I(X; \theta | \mathbf{d})$ by a specific mutual information estimator.

We consider BED in two popular models that are widely used in epidemiology and medicine studies:

- *Death process model*. In this experiment, we model the dynamics of an epidemic in which healthy individuals become infected at rate $\theta$. The design problem here is to choose observations times $\mathbf{d} \in \mathbb{R}^2$ at which to observe the number of infected individuals in two consecutive experiments, where in each experiment we observe the infection dynamics $X$ of three independent populations.

- *Pharmacokinetic model*. In this experiment, we consider finding the optimal blood sampling time $\mathbf{d} \in \mathbb{R}^M, 0 \le d_i \le 24$ for a group of patients during a pharmacokinetic (PK) study. Each $d_i$ corresponds to the sample time of each patient in the group. We would like to infer the properties of a specific drug demonstrated to the patients through the blood measurements. Here $M = 12$.

We are interested in seeing how the use of different MI estimators will affect the utility $\hat{I}(X; \boldsymbol{\theta} | \mathbf{d}^*)$ of the found design $\mathbf{d}^*$ as measured by MI. As the comparison on $\hat{I}(X; \boldsymbol{\theta} | \mathbf{d}^*)$ is only sensible when all methods estimate a lower bound of MI, we use the Donsker-Varadhan representation (2) rather than (7) to estimate MI in our method, where we set $f(x, y)$ in (2) as the log density ratio estimated by our method. This ensures that the MI estimated by our method is always a lower bound estimate.

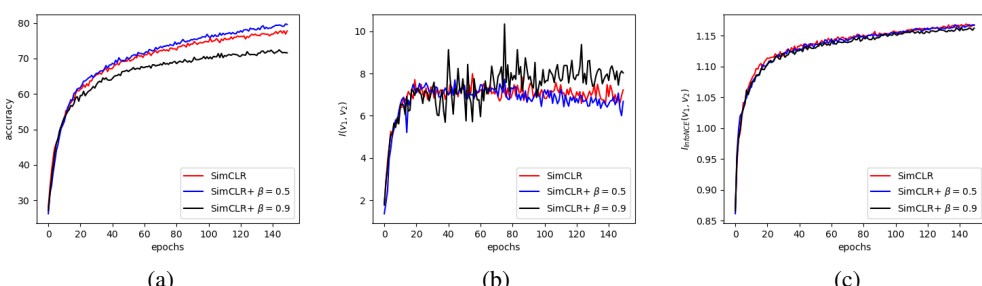

(a)                              (b)                              (c)

Figure 4: Experimental results for SSL. (a) Classification accuracy on test set as evaluated by a linear classifier. (b) $I(v_1, v_2)$ as estimated by our method. (c) $I(v_1, v_2)$ as computed by InfoNCE.

**Results**. From Figure 3.(a), it is evident that our method consistently finds a better design $\mathbf{d}^*$ for both models, especially for the PK model where the underlying MI is high. In Figure 3.(b), we further visualize the underlying function between the utility and the design in the death process model. This function is modelled by a GP trained with 300 data. As one can see, the designs found by other methods are indeed suboptimal, being away from the optimum. In Figure 3.(c), we manually inspect the design strategies reported by different methods in the PK model. Compared to other methods, the blood sampling times suggested by our method is more uniformly distributed, which is sensible as a more diverse blood sampling time will allow us to collect richer information about the drug.

### 6.3 SELF-SUPERVISED LEARNING

The third task we consider is the application of our MI estimator in self-supervised representation learning (SSL). SSL has been shown to be closely connected to infomax learning [62; 63; 64; 3; 4; 65]. For instance, [3; 4] showcase that maximizing the mutual information between different "views" of an image enables the acquisition of meaningful representations for downstream tasks. In this experiment, we aim to re-analyze SSL with our empowered estimator, investigating (a) how good can existing methods in SSL e.g. SimCLR [4] maximize the MI between different views of data; and (b) whether higher MI between views consistently results in better representations in SSL.

**Setup**. We focus on analyzing SimCLR, a classic method for SSL [4]. SimCLR learns representation by maximising the MI among various views of the same image by the InfoNCE loss [4; 2]

$$\max_{f,h} I_{\text{InfoNCE}}(h(f(v_1)), h(f(v_2)))$$

where $f$ is an encoder (implemented as a ResNet-50 model [66]) used to compute the representation of an image. $h$ is some projection function (often known as the projection head). Here the dimensionality of the representation is set to be $D = 1024$ and the batch size is set to be $B = 256$.

In addition to standard SimCLR where the estimate of MI is limited by the log batch size [67], we also consider a variant of SimCLR, SimCLR+, which can potentially attain a higher MI:

$$\max_{f,h} \beta \cdot I_{\text{MIME}}(f(v_1), f(v_2)) + (1 - \beta) \cdot I_{\text{InfoNCE}}(h(f(v_1)), h(f(v_2)))$$

Here, we use the DV representation (2) to estimate MI in our method, where we set $f(x, y)$ in (2) as the log density ratio (3) estimated in our method. This ensures that our estimate is a lower bound. We adopt the common linear evaluation protocol to assess the quality of the learned representation.

**High MI does not imply good representation**. In Figure 4, we respectively plot (a) the test set accuracy; (b) the mutual information (MI) between different views of data as estimated by our method; (c) the MI estimated by InfoNCE. These figures are collected from a typical run on CIFAR10. By comparing SimCLR and SimCLR+ with $\beta = 0.9$, we see that a high MI may not necessarily be a good sign of better representation: SimCLR+ clearly attains a higher MI but has a significantly lower test accuracy. The weak correlation between high MI and representation quality is also evidenced by the trends of the curves in Figure 4.(a) and Figure 4.(b), where the test accuracy continues to grow even when there is no increase on MI. The results echo with previous studies [62] which stated that the success of SSL may only loosely relate to MI maximization under the linear evaluation protocol.

|  | MINE | InfoNCE | MRE | MIENF | MIME |  |  | MINE | InfoNCE | MRE | MIENF | MIME |
|---|---|---|---|---|---|---|---|---|---|---|---|---|
| $\hat{I}(K;T)$ | $0.0 \pm 0.0$ | $0.1 \pm 0.0$ | $0.6 \pm 0.1$ | $0.3 \pm 0.1$ | $0.8 \pm 0.1$ | | $\hat{I}(L;R)$ | $1.3 \pm 0.2$ | $1.6 \pm 0.2$ | $1.9 \pm 0.2$ | $1.3 \pm 0.1$ | $2.8 \pm 0.1$ |
| $\hat{I}_{\text{shuff}}(K;T)$ | $0.1 \pm 0.1$ | $0.0 \pm 0.0$ | $0.1 \pm 0.0$ | $0.0 \pm 0.0$ | $0.0 \pm 0.0$ | | $\hat{I}_{\text{shuff}}(L;R)$ | $0.0 \pm 0.1$ | $0.0 \pm 0.0$ | $0.1 \pm 0.0$ | $0.0 \pm 0.0$ | $0.0 \pm 0.0$ |

(a) **kinase-target interaction**        (b) **ligand-receptor interaction**

Table 2: Quantifying dependence between participants of protein interactions. 'shuff' corresponds to cases where the data is shuffled to break dependence. Results obtained from 5 independent runs.

**InfoNCE is useful for MI maximization despite underestimation**. Another interesting result is seen by comparing Figure 4.(b) and Figure 4.(c), where we observe a significant gap between the MI estimated by our method and that by InfoNCE. As our estimate here is a lower bound of the true MI, this suggests that InfoNCE actually largely underestimates the true MI between different views of data[3]. Interestingly, despite InfoNCE largely underestimates MI, optimizing InfoNCE loss does lead to an effective maximization of the true MI (at least at the early stage of learning). This suggests that to maximize MI, an accurate estimation of its exact value is indeed unnecessary.

### 6.4 Protein sequence analysis

Our last task involves quantifying dependence in protein language model (PLM) embeddings [68]. The goal is to investigate to what extent do the PLM embeddings contain information about protein-protein interactions, which are essential for the study of their functional property. This information is quantified by the mutual information between the embeddings of different protein sequences.

**Setup**. Following the setups in [55], we consider two types of protein-protein interactions: kinase-target and ligand-receptor interactions, using annotated data from the OmniPath database [36]. For kinase-target interaction, we consider kinase and target sequence embeddings as random variables $K \in R^{1024}, T \in R^{1024}$ with joint distribution $p(K,T)$, and are interested in evaluating $I(K;T)$. Similarly, we are interested in evaluating $I(L;R)$ for ligand-receptor interactions. All data are preprocessed using the auto-encoder approach in [55] to reduce their dimensionality to 32 before MI estimation. As in previous experiments, we ensure that all MI estimates are lower bound estimates.

**Results**. For this task, our method estimates $I(K;T) \approx 0.8$nats and $I(L;R) \approx 2.8$nats. These results indicate that protein sequences as encoded by the PLM embeddings contain information about both interactions, with the information about ligand-receptor interactions much higher than kinase-target interactions — a result consistent with recent studies [55]. Methods like MINE largely underestimate this information and wrongly suggest independence for kinase-target interaction.

## 7 Conclusion

In this paper, we propose a new method for mutual information (MI) estimation in high-MI settings. A main challenge in this task is the high-discrepancy issue, which causes direct comparison between the joint distribution and the marginal distribution inaccurate. We address this issue by addition-ally comparing these distributions with their vector copula approximations, which share the same marginal distributions as the original data distribution and have Gaussian dependence structures. Experiments on diverse tasks demonstrate that such additional comparisons provide fine-grained signals for accurately modelling the true dependence in data, which leads to better MI estimate.

One limitation of the proposed method is it relies on a reasonably well-trained generative model (a vector Gaussian copula) to serve as an informed reference distribution. While this model can generally be learned effectively for structured data with moderate dimensions, it can be challenging in the case of high-dimensional, non-structured data (e.g. images). This suggests that our method may be more suited for analyzing structured data, such as embeddings of a protein language model or measurements from biological experiments, rather than directly applied to high-dimensional images. Another limitation is the additional computation cost raised by the use of generative models.

For future work, we consider exploring other choices of generative models, such as those in [38], as powerful references and baselines, as well as extending our analysis to other real-world datasets [69].

---

[3]We have included the constant $\log B$ when we compute the MI in InfoNCE (here $B$ is the batch size).

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

## A    THEORETICAL PROOFS

In this section, we are going to prove Proposition 1 and Proposition 2. Before delving into that, we first revisit the proposed MIME estimator. Specifically, MIME proposes to estimate the mutual information by learning a multinomial classifier $h$ with a set of delicate reference distributions. To train the classifier $h$, we approximate the learning objective by using $N$ samples

$$\hat{h} = \underset{h}{\operatorname{argmax}}\, \mathcal{L}_N(h) := \frac{1}{N} \sum_{i=1}^{N} \log \frac{e^{h_c(x_i,y_i)}}{\sum_{k=1}^{K} e^{h_k(x_i,y_i)}}, \quad x_i, y_i \sim p(c)p_c(x,y),$$

where we use a four-ways classifier (i.e., $K = 4$) with $p_1(x,y) = p(x,y)$ and $p_4(x,y) = p(x)p(y)$. After training, we approximate the true mutual information $I(X;Y)$ using

$$\hat{I}_{\hat{h}}^N(X;Y) := \frac{1}{N} \sum_{i=1}^{N} \hat{h}_1(x_i,y_i) - \hat{h}_4(x_i,y_i),$$

For brevity, we denote $\hat{r}(x,y) := \hat{h}_1(x,y) - \hat{h}_4(x,y)$ as the estimated ratio with the learned classifier $\hat{h}$ and $r^*(x,y) := \log \frac{p(x,y)}{p(x)p(y)}$ as the ground truth ratio. Now, we are ready to prove the consistency of the proposed MIME estimator.

### A.1    PROOF OF PROPOSITION 1

**Proposition 1** (Consistency of reference-based MI estimate)**.** *Assuming that the classifier $h_c : X \times Y \to \mathbb{R}$ is uniformly bounded. For every $\varepsilon > 0$, there exists $N(\varepsilon) \in \mathbb{N}$, such that*

$$\left| \hat{I}_{\hat{h}}^N(X;Y) - I(X;Y) \right| < \varepsilon, \forall N \geq N(\varepsilon), a.s..$$

*Proof.* Firstly, we rewrite the LHS according to our definition of $\hat{I}_{\hat{h}}^N(X;Y)$:

$$\left| \hat{I}_{\hat{h}}^N(X;Y) - I(X;Y) \right| = \left| \frac{1}{N} \sum_{i=1}^{N} \hat{r}(x_i,y_i) - I(X;Y) \right|.$$

Then, by applying the triangle inequality, we have

$$\left| \frac{1}{N} \sum_{i=1}^{N} \hat{r}(x_i,y_i) - I(X;Y) \right| \leq \left| \frac{1}{N} \sum_{i=1}^{N} \hat{r}(x_i,y_i) - \frac{1}{N} \sum_{i=1}^{N} r^*(x_i,y_i) \right| + \left| \frac{1}{N} \sum_{i=1}^{N} r^*(x_i,y_i) - I(X;Y) \right|.$$

Obviously, the second term in the RHS can be bounded by a sequence $\varepsilon_N \xrightarrow{a.s.} 0$ due to the normal strong law of large numbers under mild conditions. To bound the first term, we employ the result in [22, Appendix A], which proves that the multinomial ratio estimator is consistent. Specifically, let $\hat{h} = \operatorname{argmax}_h \mathcal{L}_N(h)$ and $\hat{r} = \hat{h}_1 - \hat{h}_4$. Then for every $\varepsilon > 0$, there exists $N \in \mathbb{N}$, such that

$$|\hat{r}(x,y) - r^*(x,y)| < \varepsilon, \forall x, y, a.s..$$

Thereby, there exists a sequence $\varepsilon_i \xrightarrow{a.s.} 0$ for each $i = 1, 2, \ldots, N$ such that

$$\left| \frac{1}{N} \sum_{i=1}^{N} \hat{r}(x_i,y_i) - \frac{1}{N} \sum_{i=1}^{N} r^*(x_i,y_i) \right| \leq \frac{1}{N} \sum_{i=1}^{N} |\hat{r}(x_i,y_i) - r^*(x,y)|$$

$$\leq \frac{1}{N} \sum_{i=1}^{N} \varepsilon_i \leq \max(\varepsilon_1, \ldots, \varepsilon_N),$$

where the first inequality holds due to the triangle inequality. Hence, for each $\epsilon > 0$, there exist an $N(\varepsilon) \in \mathbb{N}$, such that $\left| \hat{I}_{\hat{h}}^N(X;Y) - I(X;Y) \right| < \varepsilon$, almost surely. $\qquad\square$

## A.2 PROOF OF PROPOSITION 2

**Proposition 2** (Controlled error in reference-based MI estimate). *Define* $\log \hat{r}_{i,j}(x,y) := h_i(x,y) - h_j(x,y)$ *with $h$ being the classifier defined as above. Upon convergence, we have that:*

$$\left| \log \hat{r}_{1,4}(x,y) - \log \frac{p(x,y)}{p(x)p(y)} \right| \leq 3 \sup_i \left| \log \hat{r}_{i,i+1}(x,y) - \log \frac{p_i(x,y)}{p_{i+1}(x,y)} \right|,$$

*Proof.* The key of the proof is the following identify in the learned network $h$:

$$h_1 - h_4 = \sum_{i=1}^{3} (h_i - h_{i+1})$$

which, by definition,

$$\log \hat{r}_{1,4}(x,y) = \sum_{i=1}^{3} \log \hat{r}_{i,i+1}(x,y)$$

Let $r_{i,j}(x,y) = \frac{p_i(x,y)}{p_j(x,y)}$ be the true density ratio between $p_i(x,y)$ and $p_j(x,y)$. We have the following important observation:

$$\log r_{1,4}(x,y) = \sum_{i=1}^{3} \log r_{i,i+1}(x,y)$$

which is due to the fact $\frac{p_1(x,y)}{p_4(x,y)} = \frac{p_1(x,y)}{p_2(x,y)} \cdot \frac{p_2(x,y)}{p_3(x,y)} \cdot \frac{p_3(x,y)}{p_4(x,y)}$. By triangular inequality

$$\left| \sum_{i=1}^{3} \log \hat{r}_{i,i+1}(x,y) - \sum_{i=1}^{3} \log r_{i,i+1}(x,y) \right| \leq \sum_{i=1}^{3} \left| \log \hat{r}_{i,i+1}(x,y) - \log r_{i,i+1}(x,y) \right|$$

which leads to

$$\left| \log \hat{r}_{1,4}(x,y) - \log r_{1,4}(x,y) \right| \leq 3 \sup_i \left| \log \hat{r}_{i,i+1}(x,y) - \log r_{i,i+1}(x,y) \right|$$

substituting $r_1(x,y) = \frac{p(x,y)}{p(x)p(y)}$ and setting $\lambda = 3$, we have

$$\left| \log \hat{r}_{1,4}(x,y) - \log \frac{p(x,y)}{p(x)p(y)} \right| \leq \lambda \left| \log \hat{r}_{i,i+1}(x,y) - \log r_{i,i+1}(x,y) \right|, \forall i$$

which completes the proof. Note that the constant $\lambda$ is typically smaller than 3 in practice. $\square$

# B ADDITIONAL EXPERIMENTAL DETAILS AND RESULTS

## B.1 DETAILS OF THE MODELS IN SYNTHETIC TASKS

- *Swiss Roll*. In this model, data $x \in \mathbb{R}^2$, $y \in \mathbb{R}$ is generated by the following process:

$$x_1 = \frac{t \cos(t)}{21}, \qquad x_2 = \frac{t \sin(t)}{21}, \qquad y = v,$$

where

$$t = \frac{3\pi}{2}(1 + 2\Phi(\epsilon_x)), \qquad v = \Phi(\epsilon_y)$$

$$\begin{bmatrix} \epsilon_x \\ \epsilon_y \end{bmatrix} \sim \mathcal{N}\left( \begin{bmatrix} \epsilon_x \\ \epsilon_y \end{bmatrix}, \mathbf{0}, \begin{bmatrix} 1 & \rho \\ \rho & 1 \end{bmatrix} \right)$$

where $\Phi(\cdot)$ is the cumulative distribution function (CDF) of standard normal distribution. It is easy to show that this model is still in the class of non-linear transformation of Gaussian.

- *MoG 1*. This model consists of 5 equally-weighted mixture of multivariate Gaussian where the mean for the $k$th component is $\mu_k = m_k \mathbf{1}$ and the dependence coefficient in the $k$th covariance matrix is $\rho_k$. Here $m_1, ...m_5 = [-0.4, -0.1, 0, 0.1, 0.4]$ and $\rho_1, ...\rho_5 = [0.5, 0.6, 0.7, 0.8, 0.9]$.

- *MoG 2*. This model consists of 5 equally-weighted mixture of multivariate Gaussian where the mean for the $k$th component is $\mu_k = m_k \mathbf{1}$ and the dependence coefficient in the $k$th covariance matrix is $\rho_k$. Here $m_1, ...m_5 = [-0.2, -0.1, 0, 0.3, 0.4]$ and $\rho_1, ...\rho_5 = [-0.2, -0.1, 0, 0.3, 0.4]$.

## B.2 IMPLEMENTATION OF THE MAPPING $f, g$ IN VECTOR GAUSSIAN COPULA

The detailed implementation of the bijective mappings $f$ and $g$ are task-dependent. Specifically,

- **Non-representation learning tasks**. For this type of tasks, we implement the two mappings $f, g$ by a continuous flow model trained by flow matching [41]. This flow model is implemented as a 4 layer MLP with 1024 hidden units per each layer and softplus non-linearity. This MLP is trained by Adam [70] with a learning rate of $5 \times 10^{-4}$ and early stopping. We use this implementation for all the experiments in Section 6.1 (synthetic data) and Section 6.2 (Bayesian experiment design).

- **Representation learning tasks**. For this type of tasks, we implement $f, g$ as element-wise functions. In such case, the model degenerates to a Gaussian copula model. Sampling in Gaussian copula can be done by first sampling $\epsilon \sim \mathcal{N}(\epsilon; 0, \Sigma)$, then computing the rank of $\epsilon_d$ in the population: $r_d = \lceil n\Phi(\epsilon_d) \rceil$. Here $\Phi(\cdot)$ is the CDF of standard normal distribution and $n$ is the sample size of population. After that, we set $x_d$ as the $r_d$-th smallest element in the population of $X_d$. The estimation of $\Sigma$ can be done by inverting this process, which takes $\leq 30\text{ms}$ when $n = 10,000$.

## B.3 COMPUTATION TIME COMPARISON

In this section, we provide details regarding the computation time of our approach as compared to the various methods. When measuring time, we record the overall time required for an estimator to converge. Here convergence is defined as the point at which no performance improvement is observed on the validation set (with a patience period of 10 epochs). The flow model used in the proposed MIME method are constant across different tasks (but they differ in convergence time).

Table 3: Running time different MI estimators. The digits in MIME corresponds to 'time for training the generative model + time for training the critic'. Results are averaged of 5 independent runs.

| | MoG Task | Image task, original space | Image task, latent space |
|---|---|---|---|
| MINE | 140.71 | 117.31 | 107.32 |
| InfoNCE | 390.24 | 368.91 | 348.41 |
| MIENF | 127.80 | 920.21 | 553.41 |
| MRE | 40.28 | 134.21 | 100.48 |
| MIME | 125.40 + 35.47 | 915.56+96.42 | 572.57+80.58 |

### B.4 COMPARING DIFFERENT CHOICE OF REFERENCE DISTRIBUTIONS

We conducted experiments with different choices of reference distributions $q(x, y)$. The results are shown in Figure 1. Here MIME (vgc) is our original method where $q(x, y)$ is modeled by a vector Gaussian copula and hence is marginal-preserving. MIME (gc) is the case where $q(x, y)$ is modeled by a Gaussian copula, which is fast to learn from data but is less marginal-preserving. MIME (single flow) is the case where $q(x, y)$ is modelled by a single flow model. This flow is the same as in MIME (vgc). Note that this flow is likely to be more difficult to learn than the two flows $q(x)$ and $q(y)$ in VGC due to the increased dimensionality. MRE is a baseline where $q(x, y)$ is an implicit distribution that is not marginal-preserving. It is clear that MIME (vgc) achieves the best performance.

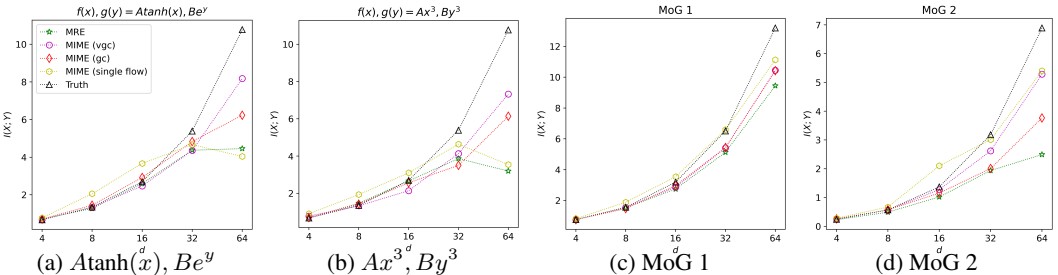

Figure 5: Comparison of different choice of reference distribution $q$. The number of training data is $N = 10,000$.

### B.5 EXPERIMENTS ON SMALL-DATA REGIME

we conducted experiments in small data regime with sample sizes $N = 2,500$ and $N = 1,250$ respectively. As shown in Figure 6 and Figure 7, the proposed estimator still demonstrates superior performance compared to the baselines in small data regimes, particularly in high-dimensional cases.

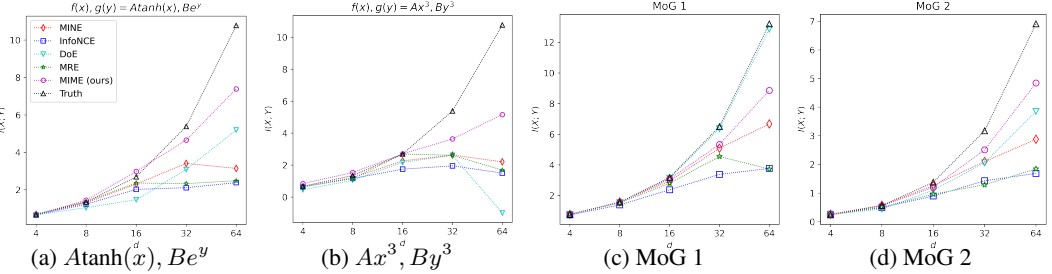

Figure 6: Comparison of different MI estimators when trained on $N = 2,500$ data.

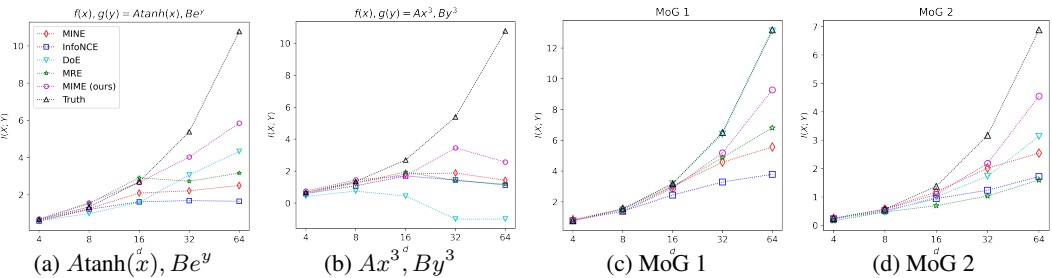

Figure 7: Comparison of different MI estimators when trained on $N = 1,250$ data.

## B.6 FURTHER RESULTS ON THE IMAGE BENCHMARKS

|  | MINE | InfoNCE | MIENF | MRE | MIME |
|---|---|---|---|---|---|
| $I(X;Y) = 3$ | $2.8 \pm 0.1$ | $2.1 \pm 0.1$ | $5.1 \pm 0.3$ | $4.2 \pm 0.7$ | $2.3 \pm 0.2$ |
| $I(X;Y) = 7$ | $5.7 \pm 0.2$ | $3.4 \pm 0.2$ | $8.7 \pm 0.3$ | $6.1 \pm 0.2$ | $3.4 \pm 0.3$ |
| $I(X;Y) = 9$ | $6.3 \pm 0.2$ | $3.9 \pm 0.2$ | $12 \pm 0.3$ | $6.4 \pm 0.3$ | $6.2 \pm 0.5$ |

(a) $\hat{I}(X;Y)$, Gaussian plots

|  | MINE | InfoNCE | MIENF | MRE | MIME |
|---|---|---|---|---|---|
| $I(X;Y) = 3$ | $2.7 \pm 0.1$ | $2.2 \pm 0.1$ | $5.1 \pm 0.3$ | $4.7 \pm 0.5$ | $2.5 \pm 0.2$ |
| $I(X;Y) = 7$ | $5.9 \pm 0.2$ | $3.7 \pm 0.2$ | $9.8 \pm 0.4$ | $6.1 \pm 0.2$ | $5.9 \pm 0.4$ |
| $I(X;Y) = 9$ | $6.2 \pm 0.2$ | $3.8 \pm 0.1$ | $11 \pm 0.4$ | $6.4 \pm 0.3$ | $6.4 \pm 0.6$ |

(b) $\hat{I}(X;Y)$, rectangles

Table 4: Experiments on the image benchmarks from [52], which contain images of correlated rectangles and Gaussian plates. MI estimation is done in the original high-dimensional space. Results are collected from 8 independent runs.

|  | MINE | InfoNCE | MIENF | MRE | MIME |
|---|---|---|---|---|---|
| $I(X;Y) = 3$ | $2.7 \pm 0.1$ | $2.3 \pm 0.1$ | $0.9 \pm 0.3$ | $3.9 \pm 0.2$ | $3.6 \pm 0.1$ |
| $I(X;Y) = 7$ | $5.4 \pm 0.2$ | $3.7 \pm 0.2$ | $1.2 \pm 0.4$ | $6.2 \pm 0.1$ | $7.5 \pm 0.4$ |
| $I(X;Y) = 9$ | $6.5 \pm 0.2$ | $3.8 \pm 0.1$ | $1.4 \pm 0.4$ | $7.0 \pm 0.4$ | $8.1 \pm 0.5$ |

(a) $\hat{I}(e(X); e(Y))$, Gaussian plots

|  | MINE | InfoNCE | MIENF | MRE | MIME |
|---|---|---|---|---|---|
| $I(X;Y) = 3$ | $2.9 \pm 0.1$ | $2.4 \pm 0.1$ | $0.7 \pm 0.1$ | $4.3 \pm 0.5$ | $3.2 \pm 0.2$ |
| $I(X;Y) = 7$ | $5.7 \pm 0.2$ | $3.7 \pm 0.1$ | $1.4 \pm 0.2$ | $6.2 \pm 0.4$ | $6.8 \pm 0.4$ |
| $I(X;Y) = 9$ | $6.3 \pm 0.2$ | $3.9 \pm 0.1$ | $2.6 \pm 0.3$ | $7.6 \pm 0.3$ | $8.3 \pm 0.4$ |

(b) $\hat{I}(e(X); e(Y))$, rectangles

Table 5: Experiments on the image benchmarks from [52], which contain images of correlated rectangles and Gaussian plots. The images are of size $32 \times 32$. We reduce the dimensionality of data by using an autoencoder $e : \mathbb{R}^{32 \times 32} \to \mathbb{R}^8$. Results are collected from 8 independent runs.

## B.7 NEURAL NETWORK ARCHITECTURE AND OPTIMIZER

**Network architecture**. For fair comparison, we use the same network architecture for the critic $f(x, y)$ in all the MI estimators considered. This network takes the form of a MLP with 3 hidden layers, each of which has 500 neurons. A densenet architecture [71] is used for the network, where we concatenate the input of the first layer (i.e., $x$ and $y$) and the representation of the penultimate layer before feeding them to the last layer. Leaky ReLU [72] is used as the activation function for all hidden layers.

**Optimizer**. We train all networks by Adam [70] with its default settings, where the learning rate is set to be $5 \times 10^{-4}$ and the batch size is set to be 512. Early stopping and a slight weight decay ($1 \times 10^{-6}$) are applied to avoid overfitting. No dropout or batch normalization is used.

