# OpenReview forum: "Neural Mutual Information Estimation with Reference Distributions"
_ICLR.cc/2025/Conference — Submitted to ICLR 2025_

### Official Review · Reviewer_MJtN · 2024-10-28

**Soundness:** 1
**Presentation:** 1
**Contribution:** 1
**Rating:** 1
**Confidence:** 5

**Summary:**

This paper considers alternate marginal-preserving base measures for the ratio-based MI estimator of Srivastava et al. Unfortunately, this paper does not provide much evidence for this choice, and the paper is dishonest in its obfuscation of its relationship to Srivastava et al., and the trivial theoretical section is apparently obfuscated to falsely appear to be more complex. While I do think the underlying idea is perhaps worth exploring, there is little in the paper as currently written that could be viewed as a contribution.

**Strengths:**

The idea of marginal preserving reference distributions might be worth exploring

**Weaknesses:**

While the idea is nice, there is little to no discussion or experiments exploring why the choice of marginal-preserving is better than the choices in [23]. The theoretical results are not related to this choice, and the experimental results are limited to hand-designed parametric distributions. The advantage in the BED results is not clear and not sufficiently explored, and the SSL results do not include [23] as a baseline and so do not contribute.

Throughout the paper, its relationship with [23] is hidden rather than discussed. The paper seems to present the idea of using reference distributions as its own (see eg title), rather than an exact copy of the framework of [23] (the only difference is the choice of reference). [23] is mentioned briefly in the related work at the end of the paper, but should be front and center in the introduction with extensive discussion motivating the change in reference distribution. Indeed, this type of discussion should have been the key contribution of this paper as it would have helped the field move forward towards better reference distributions!

The theoretical results continue this pattern of dishonesty. Proposition 1 seems to be just a direct repackaging of a result from [23], obfuscated by an additional term that is identically zero so that the triangle inequality could be invoked. Proposition 2 is trivial and does not show anything, this fact is obfuscated by the false complexity of this constant \lambda which is trivially equal to 3.

While this pattern is enough for me to not trust the experimental results, the experiments are also very lacking, mostly being tests of simple synthetic distributions. The BED experiment is not diverse or repeated, so it is impossible to tell if the method simply got lucky. Section 6.3 does not compare to [23] and therefore adds nothing.

Please use brackets [] for references to papers, not (). Using () - the same as equation references - is confusing.

Lastly, the two-step framework is not valid unless different data splits are used for the two steps. If the reference distribution depends on the samples of the distribution of interest, the samples are no longer independent and the density ratio proof does not work.

**Questions:**

see above

**Details Of Ethics Concerns:**

Throughout the paper, its relationship with [23] is hidden rather than discussed. The paper seems to present the idea of using reference distributions as its own (see eg title), rather than an exact copy of the framework of [23] (the only difference is the choice of reference). [23] is only mentioned briefly in the related work at the end of the paper, but should be front and center in the introduction with extensive discussion motivating the change in reference distribution.

The theoretical results continue this pattern of dishonesty. Proposition 1 seems to be just a direct repackaging of a result from [23], obfuscated by an additional term that is identically zero so that the triangle inequality could be invoked.

---

> ### Author Response · Authors · 2024-11-23
> **Rebuttal by the authors (1/3)**
>
> We greatly appreciate the time and effort you have dedicated to reviewing our manuscript and providing valuable feedback. While we agree with the reviewer that some aspects of the manuscript (e.g. the theory section) could be further strengthened, we strongly disagree with the serious accusation that our work raises **research integrity concerns** (e.g., plagiarism or dual submission). Our work **in no way** hides its relationship with [r1] or unnecessarily complicates any results.
>
> In the response below, we first explicitly discuss the relationship between [r1] and our work, then address each of concerns raised.
>
> **Relationship with [r1]**
>
> - *Method level:* Both our method and [r1] use reference distributions when estimating density ratios. While [r1] is designed for general-purpose density ratio estimation, our work focuses specifically on MI estimation. This specific focus allows us to design better reference distributions tailored to MI estimation, taking advantage of its marginal-irrelevant nature;
> - *Theory level:* Our work addresses two key gaps in [r1] when applied to MI estimation. First, the original consistency proof [r1] pertains to the estimated density ratio rather than the estimated MI. Second, our updated manuscript also includes an analysis on why our choice of reference distribution may be more beneficial than those in [r1] (more on this below);
> - *Experiment level:*  Our evaluation includes test cases adapted from state-of-the-art benchmarks, featuring scenarios with bounded values, skewness, high-dimensionality, varying dependence levels, coupling dimensions, long tails, and non-Gaussian dependence structures. These cases go far beyond the scenarios considered in [r1].
>
> In summary, our work is a domain-specific adaptation of [r1] with task-specific design, extended theory and enriched evaluation. We believe this was already explained in our original submission, as detailed below, but we will further clarify in our revision.
>
> **W1a. The paper is dishonest in its obfuscation of its relationship to Srivastava et al [r1].**
>
> We strongly disagree with this assessment. Below, we provide extensive evidence of how our original manuscript has sufficiently handled our relationship with [r1] with proper citation, crediting, discussion and comparison.
>
> - *Introduction and background*: in these sections, we explicitly mentioned how the research problem (high-discrepancy issue) was previously studied in [r1] and formed the motivation of our work;
> - *Methodology*: when we first introduce the reference-based MI estimator, we explicitly cited [r1] to credit the underlying technique (multi-class ratio estimation) to their work;
> - *Related works*: we spent a whole paragraph discussing the relationship between our work and [r1], stating explicitly that the use of reference distribution is not our innovation; rather, the difference to [r1] lies in the design of marginal-preserving reference distribution;
> - *Theory*: in our original manuscript, we explicitly mentioned that an important step in the proof of Proposition 1 is taken from [r1], thereby crediting this step to their work;
> - *Experiment setup*: when introducing the baseline, we again discuss directly the similarities and differences between our method and [r1]
> - *Experiment result*: in the experiment, extensive comparisons with [r1] are done to highlight the benefits of our design.
>
> Please see the anonymous document [Google document](https://docs.google.com/document/d/1lq4Gt84dkIo9OqohLsjdDJOPX4U3vdugVSRmyp4AnTc/edit?usp=sharing) here for further detailed evidence.
>
> *We note that [r1] itself is also not the first work to use reference distributions in ratio estimation; prior works such as [r3] and [r4] have explored similar methodologies. The manner in which we address our relationship to [r1] is consistent with how [r1] acknowledges and positions itself in relation to [r3] and [r4].
>
> *References*
>
> [r1] Srivastava, et. al. Estimating the density ratio between distributions with high discrepancy using multinomial logistic regression. TMLR 2023.
>
> [r2] Czyż et al. Beyond Normal: On the Evaluation of Mutual Information Estimators. NeurIPS 2023.
>
> [r3]. Nock, et.al. A scaled bregman theorem with applications, NeurIPS 2015
>
> [r4]. Bickel, et.al. Multi-task learning for hiv therapy screening. ICML 2008

---

> ### Author Response · Authors · 2024-11-23
> **Rebuttal by the authors (2/3)**
>
> **W1b.  [r1] is mentioned briefly in the related work at the end of the paper, but should be front and center in the introduction with extensive discussion motivating the change in reference distribution**
>
> We understand and appreciate the reviewer’s concern. However, we believe this primarily a choice of presentation style. Consider the following two ways of presentation:
>
> - *Big-picture first approach*. Begin with the big picture/intuition of the method *with proper citations/crediting*, followed by a detailed discussion and comparison to related works in subsequent sections.
> - *Related-work-first approach*. Start by discussing related work in depth, then introduce the newly developed methods as a way to address the limitations of prior work.
>
> Both ways are very common in the AI/ML community. Our current presentation adopt the 1st way to make the flow more accessible whereas the reviewer suggests the 2nd way.
>
> **W2.  Experiments is lacking: mostly of the tests are simple synthetic distributions; The BED experiment is not diverse or repeated; SSL experiment does not compare to [r1] and therefore adds nothing.**
>
> We address this concerns from three aspects.
>
> - *On synthetic benchmarks*. We respectfully remind the reviewer that evaluations on synthetic benchmarks are a crucial component of the modern assessment of MI estimators, as real-world dataset do not provide ground truth MI values. We also note that the synthetic benchmarks considered in our work, which are adapted and extended from state-of-the-art benchmarks [r2], are also far beyond those in [r1].
> - *BED experiments.* The results in the original BED experiments were averaged over 5 independent runs. We apologize for not including this information earlier and have explicitly clarified this in the revised manuscript;
> - *SSL experiments.* We clarify that the SSL experiment was not to compare our method with state-of-the-art estimators e.g. [r1], but to just use our method as a tool to analyze existing SSL methods;
> - *New experiments with protein data.* Finally, to strengthen our evaluation, we have also included a new experiment with real-world protein sequence data. While there is no ground truth for this dataset, we develop an evaluation protocol to faithfully compare different MI estimators in such case. Please see Section 6.4 for more details.
>
> **W3. There is little to no discussion or experiments exploring why the choice of marginal-preserving is better than the choices in [r1].**
>
> We argue that our original manuscript already included a detailed discussion on the motivation of this choice and contained extensive comparison to [r1].
>
> - In the methodology section (section 3), our original manuscript has already discussed the motivation of the use of marginal-preserving reference, which is based on the fact that mutual information is marginal-irrelevant (see the paragraph ‘Choices of reference distributions’).
> - In the experiment section (section 6), our original experiment already included extensive comparisons to [r1] on both state-of-the-art benchmarks and real-world tasks. In the appendix, we further included a comparison among different choices of reference distribution beyond those in [r1]
> - In the theory section (section 4), we have now also included an analysis (added during rebuttal) on how Proposition 2 is related to our choice of reference distribution and why it is more preferable than those in [r1].
>
> *We understand that the reviewer may want to see a direct theoretical proof of the benefit of marginal-preserving reference distributions. We argue that a method with well-explained intuition and solid empirical evaluation is still a good contribution. Just like ICLR has a long history accepting theoretic papers, it has an equally long history welcoming empirical papers.
>
>
> *References*
>
> [r1] Srivastava, et. al. Estimating the density ratio between distributions with high discrepancy using multinomial logistic regression. TMLR 2023.
>
> [r2] Czyż et al. Beyond Normal: On the Evaluation of Mutual Information Estimators. NeurIPS 2023.

---

> ### Author Response · Authors · 2024-11-23
> **Rebuttal by the authors (3/3)**
>
> **W4a.  Proposition 1 seems to be just a direct repackaging of a result from [r1], obfuscated by an additional term which is identically zero.**
>
> Thank you for this criticism. We realize that there was indeed a typo in our original proof, that one term in the triangle inequality was incorrectly stated. This resulted in the first term of the inequality being identically zero, which contributed to your impression that our proof was merely a repackaging/obfuscation of the result from [r1].
>
> We have corrected this error and sincerely apologize for the oversight. It is important to note that the triangle inequality step in our proof remains essential: while [r1] provides a consistency analysis for ratio estimation, it does not extend this analysis to mutual information (MI) estimation, missing one important step.
>
> **W4b.  Proposition 2 is trivial and does not show anything, this fact is obfuscated by the false complexity of of constant lambda.**
>
> We appreciate your criticism. We would like to address this concern from two aspects.
>
> - We acknowledge that our previous explanation of the implications of Proposition 2 may not have been sufficiently clear. We have now provided a better analysis in the revised manuscript; please see l222-l232 in Section 4. In summary, Proposition 2 says that if the references are properly chosen, so that the density ratio on the RHS can be estimated accurately by our network, the target density ratio —- the ratio of interest that suffers from the high-discrepancy issue —- will also be estimated accurately by the same network. This is specifically guaranteed by our choice of reference. Other choice of reference may not yield the same benefit;
> - We completely agree that the inequality involving \lambda can be merged into the main inequality for simplicity. We have now merged them as the reviewer requested. In the previous version, we presented them separately so as to provide readers with better insight into the actual tightness of the bound, as the two inequalities came from different sources.
>
>
> **W5. If the reference distribution depends on the samples of the distribution of interest, the samples are no longer independent and the density ratio proof does not work.**
>
> We wonder whether this is true. Taking the work [r1] as example, their reference distributions are also constructed from existing samples (especially when linear mixing strategy is used), but the estimator in [r1] still works quite well. Or do we miss something?
>
> **We welcome further comments and are happy to address any concerns.**
>
> *References*
>
> [r1] Srivastava, et. al. Estimating the density ratio between distributions with high discrepancy using multinomial logistic regression. TMLR 2023.
>
> [r2] Czyż et al. Beyond Normal: On the Evaluation of Mutual Information Estimators. NeurIPS 2023.

---

> ### Comment · Reviewer_MJtN · 2024-11-25
>
> I am disappointed by this reply, as the authors seem to have doubled down on their strategy rather than acknowledge the major flaws in how they present their work relative to prior work. Indeed, the title itself seems to claim that the paper is proposing the framework of using reference distributions for MI as their own, and the methodology section claims key preexisting equations as "our method". I suggest the authors strongly reevaluate their philosophy of the treatment of prior work, and conform to existing academic practice of clear indication at every turn whenever any component of the method is found in prior work.
>
> Regarding the theory, I see there have been some changes, but the triviality of the results have not changed, and it is deeply concerning that Proposition 1 is stated in the main text without any indication that it immediately follows from a result in Srivastava et al.
>
> Finally, the authors strangely do not seem to understand the results they are copying, as my comment on a technical limitation of the proof of one of their statements is only followed by their pointing to their misunderstanding of Srivastava et al.
>
> In sum, as the authors have failed to take these criticisms seriously, I cannot update my score and will not be replying to further comments from the authors.

---

### Official Review · Reviewer_RRtt · 2024-10-30

**Soundness:** 2
**Presentation:** 2
**Contribution:** 2
**Rating:** 5
**Confidence:** 4

**Summary:**

In this work, the so-called *high-discrepancy issue* is addressed,
This issue arises from the task of mutual information estimation between high-dimensional or highly dependent random variables,
and has been shown to target critic-based parametric mutual information estimators.
The authors suggest resolving this problem via introducing additional reference distributions to the conventional critic-based framework.
The distributions are selected to mimic the marginal distributions of the original random variables.
Throughout the work, it is argued that such approach reduces the risk of overfitting, thus mitigating the high-discrepancy issue.

**Strengths:**

1. The main idea and the underlying intuition are presented clearly.

1. The authors employ their method in several setups with different MI-related tasks,
   including plain MI estimation benchmarking on synthetic data,
   Bayesian experimental design, and self-supervised learning.
   The results indicate moderate practical advantages of the proposed technique.

**Weaknesses:**

1. The synthetic examples used to evaluate MI estimators can be considered outdated.
   A year ago, a comprehensive set of MI estimation benchmarks has been proposed in [1].
   Additionally, recent papers on MI estimation feature high-dimensional image-like datasets,
   see [2] or (20,43) from the in-paper list of references.
   Finally, there is a very recent article which also proposes new image-like datasets to benchmark MI estimators [3].
   All these tests cover many important corner cases, such as different types of long-tailed distributions,
   sparse interactions, high dimensionality, non-trivial manifolds of images.

1. Proposition 2 seems to be a direct corollary of the telescopic summation equality and the triangle inequality.
   This result does not utilize the fact that $q$ is chosen to have the same marginals as $p$.
   Thus, it gives no insights why the proposed method should work well.

1. As a consequence of the previous point, the novelty of the proposed estimator is questionable.
   Without a proper theoretical justification,
   this estimator can be seen as a version of the Multinomial Ratio Estimator with some moderate technical improvements.

1. The overview of the related works is somewhat misleading.
   Firstly, not every parametric MI estimator falls into the category of critic-based methods (lines 35--36).
   For example, MINDE estimator cited on the same lines does not employ critic networks
   and does not optimize any variational lower bound of MI (lines 37--38).

   Secondly, DV and NWJ estimators can also be viewed as the density ratio estimators (lines 73--90):
   it is known that the optimal $f$ approximated by these methods is, up to some constant,
   the density ratio in question (see section E.1 in [1]).
   The difference between DV/NWJ estimators and the cited density ratio-based approach (from the work of Hjelm et al.)
   is that in the second case, a variational bound on the Jensen–Shannon divergence is optimized
   instead of the variational bound on the KL divergence.

   Finally, the normalizing flows estimator from (43), cited on lines 236--238, is not based on density estimation.
   The technique proposed in this work allows for either a direct MI calculation
   (through the covariance matrix formula after the bi-Gaussianization)
   or a direct point-wise MI ("density ratio") calculation.
   PDF estimation is just a by-product of this method, and is not used to estimate MI.

1. Additional complexity arising from the usage of the generative models is under-addressed.
    Section B.3 should be extended to (a) include all the relevant experimental details,
    (b) provide a comprehensive analysis regarding real-life complex and high-dimensional datasets
    (such data will definitely require more sophisticated generative models to be used).

**Minor:**

1. Judging by the current state of the Bayesian experimental design section,
   it is hard to understand if the proposed estimator offers any significant practical advantages.
   The work would greatly benefit from the direct downstream performance comparison conducted for both of the tasks.
1. In equation (2), it should be mentioned that the equality holds if $\mathcal{F}$ is wide enough.
   The same goes for the NWJ bound (lines 79--80).

**Questions:**

1. What does the conditioning on $h$ in the equation (7) mean?
   From the current notation, I see that $h$ is not a random variable, but a classification (logit) function.
   The same question for similar cases of conditioning (e.g., line 365).
   Additionally, this notation seems to be inconsistent: see lines 714--716.
1. I am puzzled by the usage of the law of large numbers in Section A.1.
   It seems that the expectation and the empirical expectation (averaging) is used interchangeably (lines 741--746).
   Also, plugging $\hat I$ from lines 725--727 to the expression on lines 730--732 sets the first right-hand-side term to zero.
   I kindly ask the authors to address my concerns regarding this proof.
1. How is the time complexity in Section B.3 measured?
   It is known that InfoNCE is more resource-intensive, but it also usually requires much fewer epochs to converge.
   Are such properties fairly represented in the experimental setup?
1. I do not understand the claim on lines 80--82: as DV and NJW both approximate lower bounds on MI,
   both of them should be biased.
   I kindly ask to clarify the statement and provide a corresponding citation or proof to support it.

**Additional references:**

[1] Czyż et al. Beyond Normal: On the Evaluation of Mutual Information Estimators. *Proc. of NeurIPS 2023.*

[2] Butakov et al. Information Bottleneck Analysis of Deep Neural Networks via Lossy Compression. *Proc. of ICLR 2024.*

[3] Lee K., Rhee W. A Benchmark Suite for Evaluating Neural Mutual Information Estimators on Unstructured Datasets. *Proc. of NeurIPS 2024.*

---

> ### Author Response · Authors · 2024-11-21
> **Rebuttal by the authors (1/2)**
>
> We sincerely thank the reviewer for their valuable comments and constructive criticisms, which are truly valuable for enhancing the quality of our work. In the response below, we address your concerns regarding the weakness of our work and provide responses to your questions.
>
> **Weakness**
>
> **1. The synthetic benchmark used to evaluate MI estimators are considered outdated; there is now image-like benchmarks [r2, r3]; [r1] consider a comprehensive set of MI estimation benchmarks.**
>
> We would like to address this concern from two aspects.
>
> - *Enriched synthetic benchmarks*. We agree that we have not covered all test cases in [r1]. However, our updated test cases include representative scenarios in [r1] and also new scenarios, such as data with bounded values, skewness, low-dimensional manifold, varying dependence levels (not covered in [r1]), non-Gaussian dependence structures (not covered in [r1]) and long tails (newly added during rebuttal). The test cases in [r1, r2], by contrast, primarily exhibit a Gaussian dependence structure and have a fixed dependence level.
>
> - *Non-image real-world data*. Despite that we have not evaluated on image benchmarks, we have considered other real-world dataset in natural science, such as those from biological experiment design and protein sequence analysis (newly added during rebuttal). Although these datasets lack ground truth MI values, we have developed a protocol to compare different MI estimators in such cases (by ensuring that all estimates provided are lower bounds).
>
> Together, we believe our updated evaluation provides a comprehensive assessment. We leave evaluation on image benchmarks [r2, r3] as future work, as noted in the conclusion section.
>
> **2. The proof of Proposition 2 does not make use of the fact that the reference is marginal-preserving, thereby raising questions why the method should work well.**
>
> This is a valid concern. Let us address them from two aspects.
>
> - *Proposition 2 is still closely tied to our choice of reference distributions despite it is a general result*. Specifically, our reference distribution ensures that all terms on the RHS of Proposition 2 can be estimated accurately, so that the term on the LHS --- the target density ratio --- can also be estimated accurately. This advantage may not hold with other reference distributions. This connection is now better explained in Section 4; see l222-l232.
>
> - *Marginal-preserving references has its root in information theory*, which tells that MI is marginally irrelevant. This property suggests that a good references should encourage the network to focus on modeling differences in dependence structures rather than differences in marginals. This hypothesis is validated by our experimental comparisons with other reference choices, such as those in [r4].
>
> **3. As a consequence of the previous point, the novelty of the proposed estimator is questionable, unless there is a proper theoretical justification.**
>
> Please see our response above.
>
> In addition to the above points, we note that the work of Multinomial Ratio Estimator [r4] has not theoretically analyzed the (general) benefit of the use of reference distributions. This gap is now filled in by Proposition 2 of our work.
>
> **4. The overview of the related works is somewhat misleading.**
>
> We sincerely apologize for these inaccuracies, which have now been addressed. Specifically,
>
> - We have included a new paragraph in the background section to discuss the connection between divergence-optimizing approaches (e.g., DV, NWJ) and ratio-estimating approaches, incorporating insights from the reviewer;
> - We have revised our discussion of MINDE and [r5], correcting any misleading statements.
>
>
>
> **5. Additional computation complexity arising from the usage of the generative models is under-addressed.**
>
> We sincerely accept this criticism and promise that we will provide the full experimental details and conducted a more comprehensive analysis with real-world data soon.
>
> **Minor issues**
>
> - We fully agree that a comparison on downstream performance would be very beneficial for the BED tasks and will include it in future revision;
> - Following your suggestion, we have now mentioned that the equalities in DV bound and NWJ bound hold if $\mathcal{F}$ is wide enough.
>
> *References*
>
> [r1]. Czyż et al. Beyond Normal: On the Evaluation of Mutual Information Estimators. NeurIPS 2023.
>
> [r2]. Butakov et al. Information Bottleneck Analysis of Deep Neural Networks via Lossy Compression. ICLR 2024.
>
> [r3]. Lee K., Rhee W. A Benchmark Suite for Evaluating Neural Mutual Information Estimators on Unstructured Datasets. NeurIPS 2024.
>
> [r4]. Srivastava, et. al. Estimating the density ratio between distributions with high discrepancy using multinomial logistic regression. TMLR 2023.
>
> [r5]. Butakov et al. Mutual Information Estimation via Normalizing Flows. NeurIPS 2024.

---

> ### Author Response · Authors · 2024-11-21
> **Rebuttal by the authors (2/2)**
>
> **Questions**
>
> **1. What does the conditioning on h in the equation (7) mean? There is also some notation inconsistency in lines 714--716.**
>
> We apologize for the confusion. $\hat{I}(X;Y|h)$ meant the MI estimated with the learned network $h$. We have changed it to $\hat{I}_h(X; Y)$ to avoid confusion.
>
> The notation inconsistency has also been fixed now. Thank you very much for pointing out these issues!
>
> **2. I am puzzled by the usage of the law of large numbers in Section A.1. It seems that the expectation and the empirical expectation (averaging) is used interchangeably (lines 741--746). Also, plugging I^ from lines 725--727 to the expression on lines 730--732 sets the first right-hand-side term to zero.**
>
> Thanks for raising this concern! After careful checking, we realize that there was a typo in our original proof (one term in the triangle inequality was wrong). With this typo fixed, we believe both of your confusions have now been addressed. Please see the updated Section A.1.
>
> **3. How is the time complexity in Section B.3 measured? It is known that InfoNCE is more resource-intensive, but it also usually requires much fewer epochs to converge. Are such properties fairly represented in the experimental setup?**
>
> Thanks for this very insightful question. When measuring time complexity, we record the overall time required for an estimator to converge. Here convergence is defined as the point at which no performance improvement is observed on the validation set (with a patience period of 10 epochs). Therefore our comparison already takes into account the fast convergence property of InfoNCE. We will include this detail in our revision.
>
> **4. I do not understand the claim on lines 80--82: as DV and NJW both approximate lower bounds on MI, both of them should be biased. I kindly ask to clarify the statement and provide a corresponding citation.**
>
> Thank you for pointing out this issue! The reviewer is absolutely correct: both DV and NWJ are biased estimators. We have now corrected this statement. We sincerely apologize for the oversight in our original submission.

---

> ### Comment · Reviewer_RRtt · 2024-11-22
> **Official Comment (1/2)**
>
> I admire the work done by the authors to address my and others' concerns. In the following text, a detailed response to the official comments given by the authors is provided.
>
> ### Weaknesses
>
> 1. **Benchmarks**
>
>    I acknowledge that the authors used the Swiss Roll mapping to construct a low-dimensional manifold, and employed Gaussian mixtures to explore the non-Gaussian dependency structure. I also see the Student-t distribution in the updated version. Despite that, I still believe that the set of benchmarks used is not rich enough.
>
>    In [r1], spiral diffeomorphisms are used to construct additional tests exhibiting manifold-like structure (see Figure 5 in [r1]). Moreover, non-Gaussian tests are also considered in [r1], including the Student-t distribution (see Figure 2 or page 23 in [r1]) and a uniform distribution with additive noise (Figure 2 or page 22 in [r1]); thus, the claim "non-Gaussian dependence structures (not covered in [r1])" is incorrect.
>
>    Work [r2] suggests using even more complex manifolds of images, which resemble real data while providing tractable ground truth (GT) values of MI. In work [r3], a general method for turning a real dataset into a test with known GT MI is proposed. I believe that the source code for the benchmarks from [r1] and [r2] is publicly available, and at least one test from [r3] is easy to implement, as it is based purely on sampling random images of the same class/label.
>
>     As the work employs generative models, which are notoriously hard to train in case of high-dimensional image-like data, I insist on the importance of benchmarks from [r2,r3] (or similar) for a proper assessment of the proposed method. I believe that it is still possible to conduct at least one of the aforementioned tests before the discussion period ends.
>
> 2. **Proposition 2**
>
>    I share a general intuition with the authors regarding the reference distributions. Unfortunately, despite the provided response, I still do not consider Proposition 2 to be a rigorous justification of the proposed method. The proposition holds regardless of the choice of $ p_i $, regardless of $ h_i $ being outputs of a single NN or of separate NNs, trained using a common classification objective or separate. Trying to generalize Proposition 2 to more than two reference distributions also highlights a disconnection between the provided intuition and this theoretical result: as the number of reference distributions grows (which should result in more robust estimates), $ \lambda $ also increases, and the bound deteriorates.
>
>    To provide a rigorous theoretical justification of the method, I kindly suggest proving that $ |\hat r_{i,i+1} - r_{i,i+1}| $ can be made smaller with the right choice of the reference distributions. For example, one can try to derive a bound $ |\hat r_{i,i+1} - r_{i,i+1}| \leq C $, where $ C $ is made small by trainig the network using marginal-preserving reference distributions.
>
> 3. **Novelty**
>
>    Please, refer to the answer above. The significance of Proposition 2 in general case is limited due to the same reasoning.
>
> 4. **The overview**
>
>    This issue has been addressed properly, thank you.
>
> 5. **Computational complexity**
>
>    Thank you, I am looking forward to seeing the results. I also would like to again stress the importance of tests on image-like datasets, as such data may pose additional problems for generative models due to the increased dimensionality and complex structure.

---

> ### Comment · Reviewer_RRtt · 2024-11-22
> **Official Comment (2/2)**
>
> ### Questions
>
> 1. I am satisfied with the answer, thank you.
> 1. I have no questions regarding the proof now, thank you.
> 1. I am satisfied with the answer. However, such details should be added to the paper.
> 1. I am satisfied with the answer, thank you.
>
> ### Questions regarding the revised version
>
> I am glad to see the authors adding more experiments with real data in Section 6.4.
> However, the final conclusion in this part of the work seems questionable.
> Despite the tested methods employing lower bounds on MI, the estimated values may sometimes significantly overshoot the GT value of MI due to overfitting or other reasons (for example, see Figure 4 and Appendix C in the camera-ready version of [r5]).
> Thus, higher estimates might not indicate better estimation.
> Are such risks addressed properly in this experimental setup?
>
> ### Score
>
> I admire the noticeable effort of the authors to improve their work. All my questions have been answered, and some issues have been resolved. Despite that, I am still not able to consider this work as borderline acceptable. There is still no robust explanation why the choice of the reference distributions matters from the theoretical point of view (see the **Proposition 2** and **Novelty** weaknesses). Moreover, I am not convinced that using generative models to merely assist a critic-based method is worth it, as the gains are moderate. Additional experiments with complex datasets can be provided to address this issue (see the **Benchmarks** and **Computational complexity** weaknesses).
>
> If at least one of the two issues mentioned above is addressed, I will consider increasing my score. Moreover, if the authors provide a rigorous theoretical analysis showing the benefits from using reference distributions, I will consider this work to be complete.
>
> I would like to thank the authors again for their effort. I hope to see the aforementioned issues being resolved in the upcoming revisions.

---

> ### Author Response · Authors · 2024-11-29
> **On benchmarks, computational complexity, and real-world tasks evaluation**
>
> We deeply appreciate the reviewer’s detailed feedback and actionable suggestions. These feedback and suggestions have been extremely helpful in improving our work. In the following text, we provide responses to the reviewer’s concerns regarding benchmarks and computational complexity.
>
> **Benchmarks**
>
> Following the reviewer’s valuable suggestion, we have conducted experiments on the image-like benchmarks [r2].  Please see page 7 and Appendix B.6 of the new manuscript. We find that:
>
> - *Our generative model-based method does offer little advantage on image data.* This is due to the challenges in learning a good generative model for high-dimensional, unstructured data, which result in sub-optimal reference distributions*. We acknowledge this as a major limitation of our method, and have highlighted it in both the experimental and conclusion sections.
>
> - *Dimensionality reduction techniques help address this issue.* That said, preprocessing data with dimensionality reduction techniques, as suggested in [r2], helps significantly. By mapping the data to a low-dimensional space, generative modeling becomes much easier, leading to accurate MI estimate in our method (as well as improved estimate in other methods).
>
> *In fact, we find that the generative model-based MIENF method [r5] also does not work well in this setting. Despite our best effort, we were unable to reproduce the impressive results in [r5] with our specific implementation of flow model (a MLP trained with flow matching. This flow model is identical to that in our own method).
>
> Further analysis on the benchmarks in [r3] is on the way to assess the generality of these conclusions.
>
> **Computational complexity**
>
> We have now included a detailed report on execution time in Appendix B.3 of the revised manuscript. The report examines three different cases: (a) synthetic structured data, (b) image-like unstructured data, and (c) representations of images as computed by an autoencoder.
>
> As the reviewer predicted, generative model-based methods like ours do experience a long running time when applied to high-dimensional, unstructured data. We acknowledge this as another limitation of our method and have explicitly mentioned it in the conclusion. That said, dimensionality reduction techniques again turn out to be useful in reducing running time.
>
> **Concern regarding the evaluation on real-world dataset**
>
> We have also noticed that MI estimator can overshoot in cases of overfitting or when using a small number of Monte Carlo (MC) samples in MI calculation. To reduce the risk of overshooting, we have taken the following actions:
>
> - *Avoiding overfitting*. All estimators are trained with early stopping and regularization, and the MI is computed on the validation set.
> - *Using a large MC sample size*. We use up to 3,000 MC samples when computing the MI, which as we believe provides an accurate estimate.
>
> We would appreciate the reviewer’s insights on whether these actions are considered sufficient to address the concern of overshooting.
>
>
>
> **Summary & Discussion**
>
> At this stage, we now fully understand the reviewer’s earlier concern regarding our prior evaluation. Our method is indeed not very successful in cases involving high-dimensional, unstructured data, where the training of generative models is tricky. Although this issue can be alleviated by dimensionality reduction techniques, and it is a common issue for generative model-based methods, we recognize this as a primary limitation of our approach.
>
> The true use case for our method may be in analyzing structured data with moderate dimensions, such as model embeddings or measurements from scientific experiments. Moreover, our method may be better suited for scenarios where accuracy is prioritized over efficiency.
>
> Once again, we deeply appreciate the reviewer’s feedback and suggestions. We are eager to learn more from the reviewer’s insights. Even if the recommendation for rejection remains unchanged, your input will be instrumental for the future refinement of our work. Thank you!
>
> (the limitation regarding our theory will be discussed in another response)

---

> ### Comment · Reviewer_RRtt · 2024-11-29
> **Reply to "On benchmarks, computational complexity, and real-world tasks evaluation"**
>
> Dear authors,
>
> Thank you very much for addressing my concerns and revising your manuscript! I am now mostly satisfied with the experimental part of the work, as it is rather diverse and covers many complex and interesting cases. Although the gains are not very impressive, the results and the method, in my opinion, are still of considerable interest to the community. I will raise my score to 5.
>
> However, with the theoretical part of the work now being the main weakness, I still can not consider this submission to be acceptable (here my opinion aligns with other reviewers). Please, note that this issue is the *only* one currently keeping me from raising the score above the threshold. I sincerely hope that this problem will be resolved in the near future, as this will greatly improve the novelty of the work and will also make the experimental results more convincing.
>
> I am, of course, looking forward to the second response, which the authors mention in the final line.
>
> Below I provide some additional questions/suggestions, which are optional to consider:
> 1. In [r5], a GLOW flow model is used, which might provide some advantages in tests on image-like data. As the authors use flow matching, image-aware architectures (U-net-based, perhaps) can be considered to achieve better results.
> 2. For me, it is quite surprising that MINE outperforms InfoNCE in Tables 4 and 5, as both methods employ the same Donsker-Varadhan bound, but InfoNCE offers a better approximation for big batches. As the size of batches is reported to be $ 512 $, the "log(batch_size)" issue should not affect the estimates for $ I(X;Y) \leq 6 $. I kindly ask the authors to investigate this phenomenon. If, nevertheless, it is batch-related, please consider increasing the batch size and providing additional comparison.

---

> ### Author Response · Authors · 2024-12-02
> **On novelty and theoretic justification of the method**
>
> Thank you very much for the prompt feedback and the update on the scores, which are really encouraging. Your new suggestions regarding our recent experimental results are also extremely insightful and serve as valuable guidance in our further exploration.
>
> In the text below, we provide a more detailed analysis regarding your concerns on the novelty and the theoretic aspects of the paper, following your suggestion on analyzing how $|r_{i,i+1} - \hat{r}_{i, i+1}| < C$ is achieved.
>
> **Preparations**
>
> Rather than directly showing that $|r_{i,i+1} - \hat{r}_{i, i+1}|<C$, which is difficult, we discuss under what conditions this is likely to achieve, following the practice in the highly-cited paper [r6]:
>
> - *Small discrepancies*. The discrepancy between $p_i$ and $p_{i+1}$ is small*; or
> - *Infinite samples.* We can generate infinite samples from $p_i$ and $p_{i+1}$.
>
> We show below how our choice of reference distribution (vector Gaussian copula) satisfy these two conditions.
>
> *The original NCE paper [r7] showed that NCE-like methods work well if the two distributions are close to each other. See Theorem 3 and Section 2.4 in [r7].
>
> **A. Vector Gaussian copula is discrepancy-reducing**
>
> In [this document](https://docs.google.com/document/d/1lq4Gt84dkIo9OqohLsjdDJOPX4U3vdugVSRmyp4AnTc/edit?usp=sharing) , we show that VGC provably reduces distributional discrepancy. Specifically, let $q(x, y), q’(x, y)$ be two VGC corresponding to $p(x, y)$ and $p(x)p(y)$ respectively. Then
>
> - $q(x, y)$ must be closer to $p(x, y)$ than $p(x)p(y)$, as stated in Proposition 3 in [this document](https://docs.google.com/document/d/1lq4Gt84dkIo9OqohLsjdDJOPX4U3vdugVSRmyp4AnTc/edit?usp=sharing);
> - $q’(x, y)$  must be closer to $p(x)p(y)$ than $p(x, y)$, as stated in Proposition 4 in [this document](https://docs.google.com/document/d/1lq4Gt84dkIo9OqohLsjdDJOPX4U3vdugVSRmyp4AnTc/edit?usp=sharing).
>
> These reduced discrepancies imply that learning the ratios $p(x, y)/q(x, y)$, $q’(x, y)/p(x)p(y)$ is much easier than directly learning $p(x, y)/p(x)p(y)$. Note that during the construction of VGC, we only require its marginals to approximate the two marginals $p(x)$ and $p(y)$ well, as opposed to fitting the entire density $p(x, y)$.
>
> **B. Vector Gaussian copula allows infinite sample generation**
>
> We can generate infinite samples from the two VGC $q$ and $q’$ easily. This implies that the ratio $q(x, y)/q’(x, y)$ between the two VGC can also be learned accurately.
>
>
>
> **OK, but how about other choices of references?**
>
> From A, B above, we already see that $|r_{i,i+1} - \hat{r}_{i, i+1}|$ are small for all $i$. But do other choices of references also enjoy the same benefit as VGC? Let us consider two alternative choices:
>
> - *Other generative model.* In theory, any well-trained generative model $q(x, y) \approx p(x, y)$ can also reduce distributional discrepancies. However, this will require $q$ to approximate the entire joint density $p$ accurately, whereas VGC only requires $q$’s marginals to approximate $p$’s marginals. The latter is far more manageable (In fact, if a generative model can approximate the joint density well, it must also be able to approximate the marginals well).
> - *The implicit distributions in [r4].* Alternatively, one can use a large number of reference distributions interpolating between $p(x, y)$ and $p(x)p(y)$ to ensure small discrepancies between each $p_i$ and $p_{i+1}$. However, increasing the number of reference distributions can lead to the accumulation of estimation error, as can be seen from Proposition 2 and also noted in [r6].
>
> In Appendix B.4, we further provide a comparison to these two choices of references, which justify our choice.
>
> **Summary & Discussion**
>
> Critic-based estimators suffers from high-discrepancy issue, whereas generative estimators rely on well-trained  generative models to accurately model the distribution. We show that *even an imperfect generative model can help a lot in mitigating the high-discrepancy issue*. This generative model needs not to approximate the entire joint density well; rather, it only needs to approximate the marginal distributions. Our approach can be seen as combining the strengths of critic-based and generative methods while avoiding their respective weaknesses.
>
> Once again, we are very eager to learn from the reviewer’s insights and thoughts on our analysis, which, as always, have been instrumental in improving our work. Any criticisms and comments are welcome regardless of the rating.
>
>
> *References*
>
>
> [r4]. Srivastava, et. al. Estimating the density ratio between distributions with high discrepancy using multinomial logistic regression. TMLR 2023.
>
> [r6]. Rhode, et. al. Telescoping Density-Ratio Estimation. NeurIPS 2020.
>
> [r7]. Gutmann, et. al. Noise-contrastive estimation: A new estimation principle for unnormalized statistical models. AISTATS 2010

---

### Official Review · Reviewer_g9r3 · 2024-10-31

**Soundness:** 4
**Presentation:** 3
**Contribution:** 2
**Rating:** 5
**Confidence:** 4

**Summary:**

The authors propose a new method to address the 'high discrepancy' that neural esimtaors of mutual information suffer from (i.e., instability in high dims/ high MI regime). The authors incorporate reference distributions that  can be used to address the 'overfitting' that leads to such high discrepancy, among other phenomena. The authors present the utility and power of proposed method on several benchmarks and applications. Furthermore, the authors present several formal guarantees for their method.

**Strengths:**

- In my opinion the paper is well written and not too hard to follow.
- The presented results and experiments are easy to understand and the main contribution is overall clear.
- The experimentation is extensive and covers several benchmarks.
- an application to bayesian experiment design is presented.

**Weaknesses:**

1. Experiments - I would expect a comparison with SOTA MI estimators that were designed to deal with high dimensional data, which is known to be a caveat of neural estimation. The literature is dense. but I would choose 1-2 estimators to compare performance with (e.g. the unnormalized statistics estiamro or a normalizing flow-based estimator [1,2,3]), as the current estimators you compare to are not the best known for some time now. Such an experiment would help emphasize the advantage over current state of the art.

2. Unfortunately, I am not confident in the overall novelty of this work. The concept of reference distributions was previously explored for neural estimation [4,5]. While this work does not exactly leverage the reference distribution in the same sense as the ones referenced I would expect to see the contribution on high dimensional real world data, for which neural estimation is most crucial (when MI is just high but the data manifold is rather 'simple' both classical methods and flow-based method are expected to perform very well) beyond the weighting against the InfoNCE. One concrete example can be the design of an 'infomax-like' experiment, which shows the utility of estimated MI for representation learning, or some SSL learning which shows that MIME helps (while the current SSL experiment shows that the best approach is to consider only InfoNCE - correct me if I'm wrong).

3. I believe some technical claims can be made more rigorous and some important parts of this work can be better streamlined, see the questions section for more details.



[1] Duong, Bao, and Thin Nguyen. "Diffeomorphic information neural estimation." Proceedings of the AAAI Conference on Artificial Intelligence. Vol. 37. No. 6. 2023.‏
[2] Guo, Qing, et al. "Tight mutual information estimation with contrastive fenchel-legendre optimization." Advances in Neural Information Processing Sy
[3] Wang, Xu, et al. "Smoothed InfoNCE: Breaking the log N Curse without Overshooting." 2022 IEEE International Symposium on Information Theory (ISIT). IEEE, 2022.‏
[4] Chan, Chung, et al. "Neural entropic estimation: A faster path to mutual information estimation." arXiv preprint arXiv:1905.12957 (2019).‏
[5] Tsur, Dor, et al. "Neural estimation and optimization of directed information over continuous spaces." IEEE Transactions on Information Theory 69.8 (2023): 4777-4798.‏

**Questions:**

1. The authors discuss the approximation of the llr (proposition 2) - while we can bound the overall approximation with the maximal 'single-llr' approximation, how good do we perform a single-llr approximation?

2. Could the authors elaborate on the contribution of MIME to the SSL experiment? is the bottom line that weighting the InfoNCE with another estimator does not improve performance?

3. In equation (5), what's h_k? is h_i a logit corresponding to the ith class? this is the center of this work and I believe clearly and carefully explaining notation would significantly improve clarity.
4. what is \epsilon_{\leq d}?

5. They authors give formal guarantees (asymptotic) under the assumption of a uniformly bounded classifier. Is it guaranteed by design? can we have any conditions fo this assumption to hold?

6. 'Gaussian in the middle' (lines 211-213) - Are we guaranteed that the Gaussian copula lies between the joint and product of marginals? additionally? in what sense is the distribution between them? DKL does not have a natural geometric interpretation (actually, it does not even follow the triangle inequality..)

7. In the SSL experiment - you mention the 'true mutual information' - what is the notion of true MI in real world data stem from? do we have any mean of estimating\calculating the ground truth?


technical comments
1. I think \cal{D} definition (line 68) misses i=1 \to n notation.
2. I think a graph of the error vs n. may benefit to understand the performance of the estimator, while we don't have a non-asymptotic bound it will be good how is the convergence w.r.t. n, especially compared to other known estimators. For the y-axis, the author can consider some standard absolute error w.r.t ground truth (i.e., by consider a simple Gaussian case where ground truth is closed form).

---

> ### Author Response · Authors · 2024-11-21
> **Rebuttal by the authors (1/2)**
>
> Thank you for your detailed comments and the constructive critique, which we found truly helpful in improving the quality of our work. Below, we try to address the concerns raised as well as answering your questions.
>
> **Weaknesses**
>
> **1. Lacking comparison with state-of-the-art unnormalized and flow-based MI estimators e.g. literature [1, 2, 3]**
>
> We sincerely thank the reviewer for this valuable suggestion and the literature provided, which are now also cited and discussed in the manuscript. We have now included comparisons to state-of-the-art neural MI methods, including
>
> - *Comparison to state-of-the-art flow-based method*. Instead of directly comparing with the excellent works mentioned, we choose to compare to another work MIENF [A] in our revision, which is more recent and represents the state-of-the-art flow-based method;
>
> - *Comparison to state-of-the-art unnormalized method*. On the other hand, the baseline [B] considered in our original manuscript can already be seen as the representative of state-of-the-art of unnormalized methods. This method was tailored for high-dimensional cases and was shown to beat a wide-range of unnormalized methods.
>
> Together, we believe our current evaluation have already included comparisons with the state-of-the-arts of both flow-based methods and unnormalized methods.
>
> **2a. Novelty: the concept of reference distributions already exist in prior works; see e.g. [4, 5]**
>
> We appreciate the reviewer for sharing these important literature with us, which we should not have missed. These works have now been cited and discussed properly in the revised manuscript. Despite the concept of reference distribution is not new, we argue that our method is still novel in the following senses:
>
> - The way we use reference distributions (classifying reference distributions from the joint distribution and the product of marginals) is very different from these prior works;
> - Our design of marginal-preserving reference distribution is unique, which exploits the nature of the problem i.e. MI is marginal-irrelevant;
> - Our proposal to use generative model as informed reference distributions also establishes a new connection between unnormalized and flow-based methods.
>
> **2b. No infomax-like experiments / MIME + InfoNCE does not help in SSL**
>
> This is a valid concern, but we believe this critique can be firmly addressed by clarifying our true contribution in the SSL experiment (see our response below to your question 2).
>
> **3. Technical claims can be made more rigorous and streamlined. See the questions for more details.**
>
> We sincerely accept this critique and have improved our manuscript accordingly. Please see our responses below to your questions.
>
> *References*
>
> [A]. Butakov et. al. Mutual information estimation via normalizing flows. NeurIPS 2024.
>
> [B]. Srivastava et. al. Estimating the Density Ratio between Distributions with High Discrepancy using Multinomial Logistic Regression. TMLR 2023.

---

> ### Author Response · Authors · 2024-11-21
> **Rebuttal by the authors (2/2)**
>
> **Questions**
>
> **1. While we can bound the overall approximation with the maximal 'single-llr' approximation, how good do we perform a single-llr approximation?**
>
> This is indeed a very insightful question and touches on a key aspect of our design. In short, the accuracy of a single-LLR approximation is guaranteed by our specific choice of reference distribution (a vector Gaussian copula).  For a detailed analysis, please refer to the section immediately following Proposition 2 in the updated manuscript (page 5, l222-l232).
>
> **2. Could the authors elaborate on the contribution of MIME to the SSL experiment? It seems weighting the InfoNCE with another estimator (MIME) does not improve performance?**
>
> Many thanks for this important question. Your understanding is correct: MIME + InfoNCE indeed helps little in improving performance. However, this seemingly negative result is precisely one of our key contribution.
>
> The purpose of the MIME + InfoNCE experiment was *not* to develop a better SSL method. Rather, it was designed to explore the relationship between representation quality and the infomax principle. From our experiments, we see that MIME + InfoNCE does attain a higher MI than InfoNCE, but it actually leads to worse representation quality. This implies that representation quality is only loosely related to the infomax principle --- a key finding of our work that can contribute to better understanding current SSL methods and developing future SSL methods.
>
>
> **3. In equation (5), what's h_k? is h_i a logit corresponding to the ith class?**
>
> Completely correct. We have made this point clearer in the revised manuscript.
>
> **4. what is \epsilon_{\leq d}?**
>
>
>
> We apologize for the confusion made. It means a random variable that corresponds to the first d dimensions of \epsilon (which itself is a random variable). We have updated the text to make this definition more explicit.
>
> **5. They authors give formal guarantees (asymptotic) under the assumption of a uniformly bounded classifier. Is it guaranteed by design?**
>
> This assumption is indeed very mild and can be easily guaranteed by clipping the extreme values of the classifier’s output, though in practice we have never seen such extreme values.
>
> **6. Are we guaranteed that the (vector) Gaussian copula lies between the joint and product of marginals?**
>
> Good question! According to recent vector copula theory [C], the difference between two probabilistic distributions can be fully factorized into two parts, namely (i) the differences in the multivariate marginals; and (ii) the differences in the joint dependence structure.
>
> For a well-trained vector Gaussian copula (VGC), its marginals are nearly identical to that of the joint and the product of marginals, and its dependence structure is a Gaussian approximation of the joint dependence structure. Therefore it is very likely to be in the middle between the joint and the product of marginals.
>
> **7. What is the notion of ‘true mutual information’ in real world data stem from in the SSL experiment? Do we have any mean of estimating\calculating the ground truth?**
>
> We apologize for the confusion. What we meant was ‘the true mutual information as lower bounded by our estimator’. We have modified the statement accordingly for preciseness.
>
> When analyzing how InfoNCE underestimates the true MI, we used the gap between our method’s estimate and InfoNCE’s estimate as a proxy of the gap between the true MI and InfoNCE’s estimate, which is valid so long as our estimate is a lower bound of the true MI (see l475-l477 for how we ensure that our estimate is a lower bound estimate).
>
> **technical comments**
> - We have fixed the mentioned typo regarding the definition of \cal{D}
> - We also fully agree that a graph of the error vs n. will be highly beneficial. In the appendix, we have compared the performance of different methods with varying n.
>
>
> *References*
>
> [C]. Fan et.al. Vector copulas. Journal of Econometrics, 2023.

---

> > ### Comment · Reviewer_g9r3 · 2024-12-01
> >
> > I thank the authors for answering my questions. I will currently keep my score unchanged.
> >
> > I appreciate the discussion following the approximation statement (Prop. 2), but I don't find it too convincing. I think that Proposition 2 has concrete contribution if it is coupled with a propoer statement about the approximation capabilites of the terms that appear in the upper bound, or some quantification of how does approximation errors can be controlled, and, perhaps, under what conditions.
> >
> > Furthermore, I appreciate the authors adding a flow-based model to the benchmarking experiment. However, as it performs quite well and highly competetive with the proposed method, I would expect a propoer discussion comparing the two, and honestly oulining the advantages and disadvantages of the proposed method compared to MIENF.
> >
> > Addressing those two points will lead me to increase my score.

---

> ### Author Response · Authors · 2024-12-02
> **Thanks for the follow-up feedback; Actions made to address your new concerns**
>
> We are very grateful for your updated response and actionable feedback. Along with your previous comments, they have been highly valuable in refining our work. Below, we address the two issues raised in your follow-up response. As always, any additional insights or criticisms are welcome.
>
> **Controlled errors for terms in the upper bound in Proposition 2**
>
> We fully understand this concern and agree that our previous explanation was still not quite satisfactory. In fact, this issue is also raised by reviewer RRtt.
>
> In a new analysis, we explain detailedly how the errors in the upper bound are controlled. Rather than directly quantifying this error, which to the best of our knowledge has not been achieved in existing literature, we analyze the conditions when these errors are small, following your suggestion as well as the practice in existing works [r6, r7]. Please see the response ‘[*On novelty and theoretic justification of the method*](https://openreview.net/forum?id=fYOl9leH72&noteId=gDQ9ctqHIL)’ below to reviewer RRtt.
>
> **Comparison and discussion of MIENF**
>
> While MIENF does serve as a very strong baseline, our method still offers notable advantages in multiple cases; see e.g. Figure 1b, Figure 1c, Figure 1e and Figure 2e.
>
> Regarding the discussion comparing the two methods, our updated manuscript already includes a detailed discussion; see the last paragraph in Section 4 (this discussion was not highlighted in our last response due to the rush). In short, our method is more preferable in cases with non-Gaussian dependence structure, as discussed in Section 4 and also evidenced by Figure 2e. However, this comes at the cost of additional computation complexity, as highlighted in the conclusion of the paper.
>
>
> *References*
>
> [r6]. Rhode, et. al. Telescoping Density-Ratio Estimation. NeurIPS 2020.
>
> [r7]. Gutmann, et. al. Noise-contrastive estimation: A new estimation principle for unnormalized statistical models. AISTATS 2010

---

### Official Review · Reviewer_hcEc · 2024-11-03

**Soundness:** 3
**Presentation:** 2
**Contribution:** 2
**Rating:** 5
**Confidence:** 4

**Summary:**

The paper under review proposes a method for estimating Mutual Information (MI) based on a multinomial classification problem involving samples from a reference distribution. In a way, it can be seen as a hybrid of a discriminative and a generative approach to MI estimation. The main challenge that this paper is focused on is the so-called high-discrepancy issue, when the difference between the joint distribution and the product distribution is large. The paper included theoretical analsysis (consistency, error bounds) as well as extensive experiments comparing the new approach (named MIME) with existing methods on synthetic data and in the context of some applications.

**Strengths:**

The proposed approach is novel and very interesting. The ideas of using a reference distribution based on Gaussian copulat theory is original. The paper includes a sound consistency and error analysis and the proofs are, as far as I can tell, correct. The submission includes solid experimental comparisons with popular methods such as MINE, InfoNCE, MRE and Difference of Entropy (DoE) estimators. The experiments are applied with synthetic data and in the context of Bayesian Experimental Design and self-supervised learning. The new method shows promising results and beats some existing methods in the datasets considered. The method performs well, in particular, in addressing the high-discrepancy issue.

**Weaknesses:**

* The introduction is good, but could be improved a bit further by elaborating on the classification approach to density ratio estimation, as in Srivastava et.al.

* While the method is new and original, it is not clear from the experimental results provided to what extent it constitutes a significant improvement. Most of the experiments are just mix of Gaussians or cubed Gaussians and the results in Swiss Roll not significant. On the other hand, this is somewhat mitigated by the improved time complexity of the proposed method.

* The authors fail to reference the DEMI paper: Ruizhi Liao, Daniel Moyer, Polina Golland, and William M. Wells III. DEMI: discriminative estimator of mutual information. CoRR, 2020 This paper is a classic example of a classifier-based estimator using similar ideas.

* The paper mentions limitations of generative models such as doing well only on Gaussian mixtures. It seems that only a specific version of DoE is implemented and a new generative model, MINDE (cited in the paper) is left out of the analysis. It is not clear whether the limitations mentioned are inherent to generative models or a feature of the specific implementation of DoE used.

* The analysis on the last experiment 6.3 is somewhat limited and could be considered as known result In fact, if the mutual information estimates are non-decreasing, one could obtain similar conclusions with other methods. * InfoNCE was well analysed by Poole, 2019 and Song, 2020.

**Questions:**

In addition to addressing the above mentioned weaknesses,
* In question (5), you don't say what $K$ is.
* Could you relate your work to DEMI (mentioned above)
* A discussion of MINDE as an example of a new generative model for the sake of comparison could be helpful in assessing the merit of the method
* It would be interesting to see how the method performs with distributions such as Student t, where long-tails is known to be an issue.

---

> ### Author Response · Authors · 2024-11-21
> **Rebuttal by the authors (1/2)**
>
> Thank you for your insightful and constructive review of our manuscript. We greatly appreciate your positive remarks on our originality of work, the solid experimental comparisons, and the soundness of our consistency and error analysis. We have taken a detailed look into your questions and concerns. Below, we provide detailed responses to address these points.
>
> **Weaknesses**
>
> **1. The introduction is good, but could be improved a bit further by elaborating on the classification approach to density ratio estimation.**
>
> Thank you for the great suggestion! We have now expanded the introduction section a bit to draw a clearer connection to classification-based density ratio estimators.
>
> **2. Most of the experiments are just mix of Gaussians or cubed Gaussians and thereby not convincing enough.**
>
> Thank you for the constructive critique. We address this concern from two aspects.
>
> - *Enriched synthetic benchmarks*. Our updated test cases already include diverse scenarios, such as data with bounded values, skewness, low-dimensional manifold, varying dependence levels, non-Gaussian dependence structures (see the mix of Guassian models) and long tails (newly added during rebuttal). This set of test case already include the representative cases in state-of-the-art benchmarks [r2] and is even beyond, as [r2] has only considered Gaussian dependence structure and fixed dependence levels.
> - *Real-world dataset*. Our evaluation also include data from real-world tasks, such as Bayesian experimental design and protein sequence analysis (added during rebuttal). While ground truth MI is unavailable in these tasks, we develop a evaluation protocol for comparison different MI estimators in such cases, where we ensure all estimates are lower bounds.
>
> **3. Missing the reference DEMI**
>
> We apologize for missing this important work, which is indeed highly relevant. We have now correctly cited DEMI in both the introduction and the background section.
>
> **4. Only a specific version of DoE is implemented. Other generative model such as those in MINDE is not considered in evaluation.**
>
> This is a valid concern. We address this from two aspects.
>
> - *Our current choice of generative models is for controlled comparison*. Specifically, we chose this particular implementation of DoE to make sure that the generative models used in both DoE and our method are the exactly same, so as to highlight the benefits brought by our estimation strategy in evaluation.
>
> - *Comparison to state-of-the-art generative model-based methods*. That being said, following the reviewer’s valuable suggestion, we have now compared to a very recent generative model-based method [r1], with MINDE also mentioned and discussed. The results confirm the advantage of our method even compared to this strong baseline.
>
>
> **5. The analysis on 6.3 may be known: if the mutual information estimates are non-decreasing, one could obtain similar conclusions with other methods.**
>
> We agree that similar conclusions may have been *qualitatively* discussed in existing works. However, our contribution lies in *quantifying* the actual gap between the true mutual information (as lower-bounded by our powerful estimator) and InfoNCE’s estimate, which is a more rigorous analysis. With this quantitative analysis, we find that method that *largely* underestimates MI can still stably maximizes MI and produces high-quality representations. This discovery may not be possible to reveal by other estimators not tailored for detecting high MI.
>
>
>
> *References*
>
> [r1]. Butakov, et.al. Mutual information estimation via normalizing flows. NeurIPS 2024.
>
> [r2]. Czyż et. al. Beyond normal: On the evaluation of mutual information estimators. NeurIPS 2023.

---

> ### Author Response · Authors · 2024-11-21
> **Rebuttal by the authors (2/2)**
>
> **Questions**
>
> - *In question (5), you don't say what K is*: thanks for pointing out this! We have fixed this issue.
> - *Can we relate to DEMI?* We have now explicitly relate our method to DEMI in introduction and background as the reviewer kindly suggested.
> - *A discussion of MINDE as an example of a new generative model for the sake of comparison:*  This is a valuable suggestion. While we have not compared to this specific MINDE method, which we have now also cited and discussed, we have compared to a more recent generative model-based method [r1]. We hope these efforts useful in addressing your concerns.
> - *Experiment on long-tails distributions*: following the reviewer’s valuable suggestion, we have now included a new experiment on (multivariate) Student-t distribution. The results again highlight the effectiveness and robustness of our method.

---

### Official Review · Reviewer_h51p · 2024-11-06

**Soundness:** 3
**Presentation:** 3
**Contribution:** 2
**Rating:** 5
**Confidence:** 4

**Summary:**

The paper presents a novel method, MIME (Mutual Information Multinomial Estimation), for estimating mutual information (MI) in high-dimensional data, addressing challenges in traditional MI estimators, especially in cases of high mutual information or high data dimensionality.

MIME introduces "reference distributions" that mimic the marginal distributions of original data while differing in dependency structures, helping to mitigate overfitting by offering more fine-grained signals for modeling dependencies. MIME formulates MI estimation as a multinomial classification problem, where a classifier is trained to distinguish between samples from the joint distribution, product of marginals, and additional reference distributions.

The authors validate MIME's effectiveness with experiments on synthetic non-Gaussian datasets and real-world tasks like Bayesian experimental design and self-supervised learning, showing MIME's resilience in complex dependency settings where traditional methods often overfit.

**Strengths:**

The methodology is novel and the problem considered in the work is important for the community.

The MIME  approach demonstrates key strengths, particularly in its innovative handling of high-dimensional and high-MI data, where traditional estimators often struggle due to overfitting. By incorporating reference distributions with similar marginals but varied dependency structures, MIME provides a more nuanced, stable estimation, which is essential in scenarios prone to high discrepancy issues.

The method’s design as a multinomial classification problem with multiple reference points is an innovative strategy to alleviate the challenges seen in conventional approaches, such as overfitting and limited robustness.

**Weaknesses:**

Something important which is missing is the discussion about whether the proposed method is or not a bound of the MI. If it is a bound, while Proposition 1 is correct (see comments later), it does not provide intuition about the eq.(4) problem.
It is not immediately clear that the statement in proposition 2 constitutes in practice an advantage.

One further drawback is the high computational complexity of the proposed method, which combines generative models and discriminators.

The work lacks in terms of experimental validation. Given the structural similarity between this work and [43], it would be interesting to see a comparison on the performance. Also, I would include other techniques like the one described in [20].
Moreover, the number of experiments on synthetic data is rather limited, for example in the benchmark [49] there are tens of distributions.

Concerning the experiments in Section 6.2 I think that showing the bound on MI is not conclusive. Being there no ground truth, it is not possible to claim that one method is better than an other (what if MIME is over estimating the real MI?) I think in such a simple scenario it would be better to compare estimators of $\theta$ given the dataset X under the different policies, which is ultimately what we care about.

The limitation in the number of competitors also affect the results in Section 6.3. There are many different neural estimators which could have been used of the maximization of MI between views, showing that MIME (with beta=0.5) outperforms InfoNCE is not sufficient to claim that the method is robust.

**Questions:**

Can the authors expand the experimental validation as described in weaknesses?

The authors should better clarify the relationship between the technique of eq. (8) and [43], where a mutual information estimator based on normalizing flows is proposed. Can the methods be combined? What are the pros of the (seemingly) more complex procedure in which a flow model and a discriminator is learned?

Can the authors explain how is it possible that in Table 1 (appendix B.3) there is such a huge difference between the running time of DoE and MIME?

I suggest the authors to include some extra details (or assumptions) to make sure that the claim in line 734 is correct (the law of large numbers applies only in certain conditions). Also, is it a typo the \hat{h}_t in line 737?

Is the footnote 1) also containing a typo? (f=log(p(x,y))p(x)p(y)). If this is not the case I don’t understand what is written in lines 406-407.

Please clarify line 308 and the relationship with (8)

Is the proposed method a bound of the true MI? Please specify if the result of eq(4) affects the estimate and why the proposed method is not affected by log(B) (cfr [69] and the statement in line 447)

---

> ### Author Response · Authors · 2024-11-20
> **Rebuttal by the authors (1/2)**
>
> Many thanks for your constructive feedback and insightful comments. Your acknowledgements on the novelty and the strength of the work are highly encouraging, and we learn a lot from your constructive critique and suggestions. These criticisms will be addressed comprehensively in our responses below.
>
> **Weaknesses**
>
>
> **1. Whether the proposed method is a bound of MI or not is unclear.**
>
> Thank you for this important question! The answer actually depends on the equation we use in MI estimation. Specifically, once we have learned the network h (which itself estimates the log density ratio), we can calculate MI in two ways:
>
> - *Computing MI as the averaged log density ratio (eq.7)*. This is our default choice. In this case, the estimated MI is not a bound. We use this estimator in the synthetic experiments;
> - *Computing MI using the DV representation (eq.2)*. This is our optional choice. In this case, the estimated MI is a lower bound of MI. We use this estimator in all the real-world experiments.
>
> This information, along with the guidance of which way to use in practice, was briefly mentioned in footnote 1 in the original manuscript and is now further highlighted for clarity.
>
>
> **2. Proposition 1 does not provide intuition about the eq.(4) problem (the high-discrepancy issue). It is not immediately clear that the statement in proposition 2 constitutes in practice an advantage.**
>
> - Regarding Proposition 1, we clarify that it serves solely as a consistency analysis and is not intended to address the high-discrepancy issue defined in eq.(4) (it is Proposition 2's job).
> - Regarding Proposition 2, we admit that the previous presentation did not provide a clear explanation on its practical implication. We have now included a clearer explanation; see the highlighted text after Proposition 2 (page 5, l222-l232). In short, Proposition 2 implies that if the reference distributions are well-chosen, so that the ratios on the RHS of Proposition 2 will be estimated accurately, then ratio on the LHS —- the ratio of interest that suffers from the high-discrepancy issue —- can also be estimated accurately.
>
>
> **3. The work lacks in terms of experimental validation. It lacks comparison to [43] which has a similar structure as the proposed method. The number of experiments on synthetic data is rather limited (in the benchmark [49] there are tens of distributions).**
>
> We accept this criticism and have conducted additional experiments to address this issue.
>
> - *New comparison to [43]*: Following the reviewer’s valuable suggestion, we have conducted a comparison to [43], which indeed serves as a strong baseline. Despite its strength, our method still demonstrates comparable or even superior performance, particularly in cases with non-Gaussian dependence structures, such as mixtures of multivariate Gaussians.
>
> - *Enriched synthetic benchmarks*:  We agree that we have not covered all test cases in [49].  However, our updated test cases include representative scenarios in [49] and beyond, such as data with bounded values, skewness, low-dimensional manifold, varying dependence levels (not covered in [49]), non-Gaussian dependence (not covered in [49]) and long tails (newly added during rebuttal). The test cases in [49], by contrast, primarily exhibit a Gaussian dependence structure and have a fixed dependence level.
>
>
> **4. There is no ground truth in the real-world experiments in Section 6.2, so it is not possible to claim that one method is better than an other.**
>
> We are on the same page: we have also noticed there is no ground truth in real-world experiments. However, it is still possible to compare different MI estimators without ground truth, so long as all estimates are lower bounds estimates. This is realized by leveraging the DV representation of MI (see our response above). We have highlighted this information in Sec 6.2.
>
>
> **5. Showing that MIME (with beta=0.5) outperforms InfoNCE is not sufficient to claim that the method is robust.**
>
> We totally agree. In fact, other choices of beta (e.g. beta=0.9) actually leads to a worse performance. However, this seemingly negative result is precisely one of our key contribution.
>
> The primary goal of our SSL experiment was not to develop a better SSL method, but to explore the *relationship between representation quality and the infomax principle*. Through our experiments, we find that method attaining higher MI (e.g. MIME with beta=0.9) does not necessarily outperform those attaining lower MI (e.g. MIME with beta=0.5 or InfoNCE alone), suggesting that representation quality is only loosely related to the infomax principle —- a key discovery in our work.

---

> ### Author Response · Authors · 2024-11-20
> **Rebuttal by the author (2/2)**
>
> **Questions**
>
> **1. Can you expand the experimental validation as described in weaknesses**
>
> We have conducted the suggested experiments, including comparison to [43] and including new test cases in synthetic tasks. To better convince the reviewer, we have further included a new real-world task with protein sequence data.
>
> **2. Clarifying the relationship between [43] and eq.8 / Why our two-stages method is more beneficial**
>
> Thanks for this great suggestion! We have now included a discussion on the relationship between [43] and the vector Gaussian copula (VGC) model defined in eq.8 in related works.
>
> In short, the work [43] can be seen as modeling the data distribution as a VGC and directly computing MI from this model. This way of MI estimation is accurate if the underlying distribution is VGC-like, however it may be inaccurate in the case of model mis-specification. This is why our two-stages method is beneficial: instead of fully relying on the VGC model, which can be inaccurate due to model mis-specification, our method only uses it as a high-quality reference, which is safer. The experimental comparison with [43] verified this hypothesis.
>
> **3. Why there is huge difference between the computation time of DoE and MIME**
>
> This is mainly because DoE requires to fit a complex conditional distribution p(y|x), whereas our method only fits two low-dimensional distributions p(x) and p(y), which converge much faster. Note that the time spent on classifier training itself is very small compared to density estimate.
>
> **4. Extra details (or assumptions) to make sure that the claim in line 734 is correct (the law of large numbers applies only in certain conditions)**
>
> We thank the reviewer for the reminder and have added the corresponding details (i.i.d samples and finite expectation). These conditions are very mild and can be easily met in our method.
>
> **5. Line 308 and the relationship with (8)**
>
> They actually correspond to the same distribution family (vector Gaussian copula). We will clarify this in revision.
>
> **6. Why the proposed method is not affected by log(B) like InfoNCE?**
>
> This is a very insightful question. While both our method and InfoNCE works by K-way classification, the situation is completely different in the two methods. In InfoNCE, they classify one sample from other samples in the same mini-batch, whereas in our method, we classify between samples from different distributions. The unique way that InfoNCE classifies samples is the main reason that it suffers from the log(B) issue.
>
> **7. Please specify if the result of eq(4) affects the estimate of our method.**
>
> Our method is designed to sidestep the eq(4) problem (curse of high-discrepancy). This is a joint effect of (a) the proposed reference-based MI estimation procedure; (b) our specific choice of reference distribution. Please Proposition 2 and the explanation right after it in the updated manuscript for the detailed analysis.
>
> Other issues/questions such as typos and assumptions have been fixed. We sincerely thank the reviewer for pointing out them!

---

> > ### Comment · Reviewer_h51p · 2024-12-01
> >
> > I appreciate the authors' commendable efforts and the improvements made to the paper. However, given the numerous concerns raised by both myself and other reviewers, as well as the absence of experimental comparisons with MINDE, I have decided to maintain my original score.

---

> ### Author Response · Authors · 2024-12-02
> **Many thanks for reading our rebuttal; one comment regarding MINDE**
>
> We are very thankful for the time you spent reading our rebuttal, and we can understand your decision. We would be grateful if you could provide additional feedback on which concerns remain unaddressed. Regardless of the final decision, your feedback will be essential as we continue to refine our work.
>
> *On comparison to MINDE*: Although we did not directly compare to MINDE due to space constraints, we have made comparisons to a more recent baseline [r1], as kindly suggested by you and other reviewer (note that [r1] has not directly compared to MINDE either). Furthermore, our baselines encompass all the neural network-based baselines considered in both [r1] and MINDE and extend beyond (we consider up to 6 different methods). We hope our choices of baselines understandable given the space limit and the diversity of the current baselines.
>
> A concern regarding research integrity was raised by one reviewer, which is completely unfounded (see our response to this reviewer). We hope this has not influenced your decision.
>
> Thank you once again for reviewing our work and providing such valuable feedback!
>
> [r1]. Butakov et al. Mutual Information Estimation via Normalizing Flows. NeurIPS 2024.

---

### Meta-Review · Area_Chair_KRCb · 2024-12-20

**Metareview:**

This paper addresses the challenge of estimating mutual information (MI), particularly the difficulty of estimation in high-dimensional data or data with strong dependencies (the so-called "high disparity" problem), by proposing a novel approach that leverages reference distributions. The method introduces reference distributions with differing dependency structures while preserving the marginal distributions of the original random variables, aiming to improve estimation accuracy.
However, as noted by most reviewers, the specific advantages and the justification for the selection of reference distributions are insufficiently discussed. Additionally, the experimental evaluation lacks systematic comparisons with state-of-the-art (SOTA) methods, such as MINDE, and other prominent approaches. Furthermore, theoretical aspects, including their implications and validity, have been questioned by reviewers. Given these shortcomings, while the research direction is interesting, substantial revisions are required for publication. Therefore, I recommend rejection at this time.

**Additional Comments On Reviewer Discussion:**

Many reviewers pointed out the lack of comparisons with existing methods. Although additional experiments were included during the discussion phase, comparisons with certain methods, such as those highlighted by Reviewer h51p, are still missing. Reviewer RRtt raised concerns about the theoretical aspects, and additional discussions were provided to address these issues. While this partially resolved the concerns, the theoretical novelty remains weak. If the strength of this paper lies in its empirical insights, the current numerical evaluations are insufficient to support its claims.

---

### Decision · Program_Chairs · 2025-01-22

Reject